# MorphGrower: A Synchronized Layer-by-layer Growing Approach for Plausible and Diverse Neuronal Morphology Generation

## Abstract

Neuronal morphology is essential for studying brain functioning and understanding neurodegenerative disorders, e.g. Alzheimer. As the acquiring of real-world morphology data is expensive, computational approaches especially learning-based ones e.g. MorphVAE for morphology generation were recently studied, which are often conducted in a way of randomly augmenting a given authentic morphology to achieve both plausibility and diversity. Under such a setting, this paper proposes **MorphGrower** which aims to generate more plausible morphology samples by mimicking the natural growth mechanism instead of a one-shot treatment as done in MorphVAE. In particular, MorphGrower generates morphologies layer by layer synchronously and chooses a pair of sibling branches as the basic generation block, and the generation of each layer is conditioned on the morphological structure of previous layers and then generate morphologies via a conditional variational autoencoder with spherical latent space. Extensive experimental results on four real-world datasets demonstrate that MorphGrower outperforms MorphVAE by a notable margin. Our code will be publicly available to facilitate future research.

## 1 Introduction and Related Works

Neurons are the building blocks of the nervous system and constitute the computational units of the brain (Yuste, 2015). The morphology of neurons plays a central role in brain functioning (Torben-Nielsen & De Schutter, 2014) and has been a standing research area dating back to a century ago (Cajal, 1911). The morphology determines which spatial domain can be reached for a certain neuron, governing the connectivity of the neuronal circuits (Memelli et al., 2013).

A mammalian brain typically has billions of neurons (Kandel et al., 2000), and the computational description and investigation of the brain functioning require a rich number of neuronal morphologies (Lin et al., 2018). However, it can be very expensive to collect quality neuronal morphologies due to its complex collecting procedure with three key steps (Parekh & Ascoli, 2013): *i)* histological preparation, *ii)* microscopic visualization, and *iii)* accurate tracing. Such a procedure is known to be labor-intensive, time-consuming and potentially subject to human bias and error (Schmitz et al., 2011; Choromanska et al., 2012; Yang et al., 2020).

Efforts have been made to generate plausible neuronal morphologies by computational approaches, which are still immature. Traditional neuronal morphology generation methods which mainly aim for generation from scratch, can be classified into two main categories: sampling based (Samsonovich & Ascoli, 2005; da Fontoura Costa & Coelho, 2005; Farhoodi & Kording, 2018) and growth-rule based (Shinbrot, 2006; Krottje & Ooyen, 2007; Stepanyants et al., 2008) methods. Sampling-based methods do not consider the biological processes underlying neural growth (Zubler & Douglas, 2009). They often generate from a simple morphology and iteratively make small changes via e.g. Markov-Chain Monte-Carlo. They suffer from high computation overhead. As for the growth-rule-based methods, they simulate the neuronal growth process according to some preset rules or priors (Kanari et al., 2022) which can be sensitive to hyperparameters.

Beyond the above traditional methods, until very recently, learning-based methods start to shed light on this field MorphVAE (Laturnus & Berens, 2021). Besides its technical distinction to the above non-learning traditional methods, it also advocates a useful generation setting that motivates our work: generating new morphologies which are similar to a given morphology while with some (random) difference akin to the concept of data augmentation in machine learning. The rationale and value of this 'augmentation' task are that many morphologies could be similar to each other due to their same gene expression (Sorensen et al., 2015; Paulsen et al., 2022; Janssen & Budd, 2022), located in the same brain region etc (Oh et al., 2014; Economo et al., 2018; Winnubst et al., 2019). Moreover,

generation-based on a given reference could be easier and has the potential of requiring less human interference, and the augmentation step could be combined with and enhance the 'from scratch generation' as a post-processing step where the reference is generated by the traditional methods[1].

Specifically, MorphVAE proposes to generate morphologies from given samples and use those generated data to augment the existing neuronal morphology database. It defines the path from soma to a tip (see detailed definitions of soma and tip in Sec. 2) as a 3D-walk and uses a sequence-to-sequence variational autoencoder to generate those 3D-walks. MorphVAE adopts a post-hoc clustering method on the generated 3D-walks to aggregate some nodes of different 3D-walks. This way, MorphVAE can construct nodes with two or more outgoing edges and obtain a final generated morphology. Meanwhile, MorphVAE has limitations. It chooses the 3D-walk as a basic generation block. Yet 3D-walks usually span a wide range in 3D space, which means each 3D-walk is composed of many nodes in most cases. Such a long sequence of node 3D coordinates is difficult to generate via a seq2seq VAE. In fact, little domain-specific prior knowledge is incorporated into the model design, e.g., the impacts on subsequent branches exerted by previous branches. Moreover, after the post-hoc clustering, there may exist other nodes that have more than two outgoing edges in the final generated morphology apart from the soma. This violates the fundamental rule that soma is the only node allowed to have more than two outgoing branches (Bray, 1973; Wessells & Nuttall, 1978; Szebenyi et al., 1998). That is to say that morphologies generated by MorphVAE may be invalid in terms of topology (see an example in Appendix C.1).

In contrast to the one-shot generation scheme of MorphVAE, we resort to a progressive way to generate the neuronal morphologies for three reasons: 1) the growth of a real neuronal morphology itself needs one or more weeks from an initial soma (Budday et al., 2015; Sorrells et al., 2018; Sarnat, 2023). There may exist dependency over time or even causality (Pelt & Uylings, 2003). Hence mimicking such dynamics could be of benefit; 2) the domain knowledge is also often associated with the growth pattern and can be more easily infused in a progressive generation model (Harrison, 1910; Scott & Luo, 2001; Tamariz & Varela-Echavarría, 2015; Shi et al., 2022); 3) generating the whole morphology is complex while it can be much easier to generate a local part e.g. a layer in each step.

Accordingly, we propose a novel method entitled MorphGrower, which generates high-quality neuronal morphologies at a finer granularity than MorphVAE to augment the available morphology database. Specifically, we adopt a layer-by-layer generation strategy (which in this paper is simplified as layer-wise synchronized though in reality the exact dynamics can be a bit asynchronous) and choose to generate branches in pairs at each layer. In this way, our method can strictly adhere to the rule that only soma can have more than two outgoing branches throughout the generation process, thus ensuring the topological validity of generated morphologies. Moreover, based on the key observation that previous branches have an influence on subsequent branches, which has been shown in extensive works (Burke et al., 1992; Van Pelt et al., 1997; Cuntz et al., 2007; Purohit & Smith, 2016), we frame each generation step to be conditioned on the intermediate generated morphology during the inference. **The highlights of the paper (with code publicly available) are:**

1) To handle the complexity of morphology (given the authentic morphology as reference), mimicking its biologically growing nature, we devise a layer-by-layer conditional morphology generation scheme called **MorphGrower**. To our best knowledge, this is the first morphological plausibility-aware deep model for morphology generation, particularly in contrast to the peer MorphVAE which generates the whole morphology in one shot and lacks an explicit mechanism to respect the topology validity.

2) Quantitative and qualitative results on real-world datasets show that MorphGrower can generate more diverse and plausible morphologies than MorphVAE. As a side product, it can also generate snapshots of morphologies in different growing stages. We believe the computational recovery of dynamic growth procedure would be an interesting future direction which is currently rarely studied.

## 2 PRELIMINARIES

A neuronal morphology can be represented as a directed tree (a special graph) $T = (V, E)$. $V = \{v_i\}_{i=1}^{N_v}$ denotes the set of nodes, where $v_i \in \mathbb{R}^3$ are the 3D coordinates of nodes and $N_v$ represents the number of nodes. $E$ denotes the set of directed edges in $T$. An element $e_{i,j} \in E$ represents a

---

[1]As from-scratch generation quality is still far from reality without experts' careful finetuning (Kanari et al., 2022), so it is non-trivial for first scratch generation then post augmentation which we leave for future work.

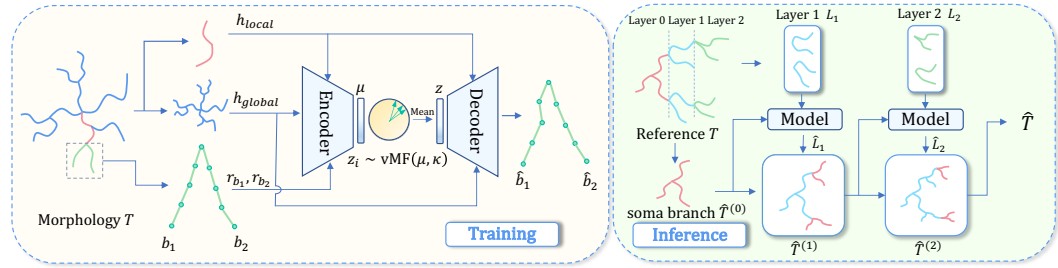

Figure 2: **Overview of MorphGrower.** It takes the branch pair and its previous layers as inputs. The branch pairs as well as the previous layers as conditions determine the mean direction $\mu$ of latent space which follows a von-Mises Fisher distribution with fixed variance $\kappa$. Latent variables are then sampled from the distribution $\mathbf{z}_i \sim \mathrm{vMF}(\mu, \kappa)$. Finally, the decoder reconstructs the branch pairs from the latent variable $\mathbf{z}$ and the given condition. In inference, the model is called regressively, taking the generated subtree $\hat{T}^{(i)}$ and the $(i+1)$-th layer $L_{i+1}$ of the reference morphology as input and outputting a new layer $\hat{L}_{i+1}$ of the final generated morphology.

directed edge, which starts from $v_i$ and ends with $v_j$. The definitions of key terms used in this paper or within the scope of neuronal morphology are presented as follows:

**Definition 1** (**Soma & Tip & Bifurcation**). Given a neuronal morphology $T$, *soma* is the root node of $T$ and *tips* are those leaf nodes. Soma is the only node that is allowed to have more than two outgoing edges, while tips have no outgoing edge. There is another special kind of nodes called *bifurcations*. Each bifurcation has two outgoing edges. Soma and bifurcation are two disjoint classes and are collectively referred to as *multifurcation*.

**Definition 2** (**Compartment & Branch**). Given a neuronal morphology $T = (V, E)$, each element $e_{i,j} \in E$ is also named as *compartment*. A *branch* is defined as a directed path, e.g. in the form of $e_{i,j} \rightarrow e_{j,k} \rightarrow \ldots e_{p,q} \rightarrow e_{q,s}$, which means that this branch starts from $v_i$ and ends at $v_s$. We can abbreviate such a branch as $b_{is}$. The beginning node $v_i$ must be a multifurcation (soma or bifurcation) and the ending node must be a multifurcation or a tip. Note that there is no other bifurcation on a branch apart from its two ends. Especially, those branches starting from soma are called *soma branches* and constitute the *soma branch layer*. Besides, we define a pair of branches starting from the same bifurcation as *sibling branches*.

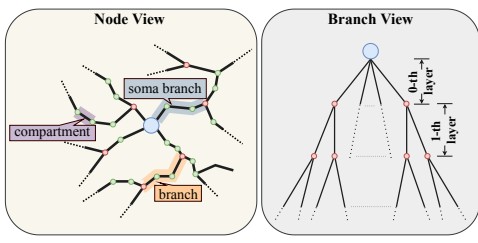

Figure 1: Neuronal morphology in node and branch views. Blue node represents soma and pink nodes represent bifurcations.

To facilitate understanding of the above definitions, we provide a node-view topological demonstration of neuronal morphology in the left part of Figure 1. Based on the concept of the branch, as shown in the right part of Figure 1, a neuronal morphology $T$ now can be decomposed into a set of branches and re-represented by a set of branches which are arranged in a Breadth First Search (BFS) like manner, i.e. $T = \{\{b_1^0, b_2^0, \ldots, b_{N^0}^0\}, \{b_1^1, b_2^1, \ldots, b_{N^1}^1\}, \ldots\}$. $b_i^k$ denotes the $i$-th branch at the $k$-th layer and $N^k$ is the number of branches at the $k$-th layer. We specify the soma branch layer as the 0-th layer.

## 3 METHODOLOGY

In this section, we propose MorphGrower to generate diverse high-quality realistic-looking neuronal morphologies. We highlight our model in Figure 2.

### 3.1 GENERATION PROCEDURE FORMULATION

**Layer-by-layer Generation Strategy.** The number of nodes and branches varies among neuronal morphologies and there exists complex dependency among nodes and edges, making generating $T$ all at once directly challenging. As previously noted, we can represent a morphology via a set of branches and these branches can be divided into different groups according to their corresponding layer number. This implies a feasible solution that we could resort to generating a morphology layer by layer. This layer-by-layer generation strategy is consistent with the natural growth pattern of

neurons. In practice, dendrites or axons grow from soma progressively and may diverge several times in the growing process (Harrison, 1910; Scott & Luo, 2001; Tamariz & Varela-Echavarría, 2015; Shi et al., 2022). In contrast, MorphVAE (Laturnus & Berens, 2021) defines the path from soma to tip as a 3D-walk and generates 3D-walks one by one. This strategy fails to consider bifurcations along the 3D-walks. Thus it violates this natural pattern.

Following such a layer-by-layer strategy, a new morphology can be obtained by generating new layers and merging them to intermediate generated morphology regressively. In the following, we further discuss the details of how to generate a certain layer next.

**Generate Branches in Pairs.** Since the leaf nodes of an intermediate morphology are either bifurcations or tips, branches grow in pairs at each layer except the soma branch layer, implying that $N^i$ is even for $i \geq 1$. As pointed out in previous works (Uylings & Smit, 1975; Kim et al., 2012; Bird & Cuntz, 2016; Otopalik et al., 2017), there exists a complex dependency between sibling branches. If we separate sibling branches from each other and generate each of them individually, this dependency will be hard to model. A natural idea is that one can regard sibling branches as a whole and generate sibling branches in pairs each time, to implicitly model their internal dependency. Following Laturnus & Berens (2021), we adopt a seq2seq variational autoencoder (VAE) for generation. Different from the work by Laturnus & Berens (2021), which uses a VAE to generate 3D-walks with greater lengths in 3D space than branches, increasing the difficulty of generation, our branch-pair-based method can generate high-quality neuronal morphologies more easily.

**Conditional Generation.** In addition, a body of literature on neuronal morphology (Burke et al., 1992; Van Pelt et al., 1997; Cuntz et al., 2007; Purohit & Smith, 2016) has consistently shown such an observation in neuronal growth: grown branches could influence their subsequent branches. Hence, taking the structural information of the first $i$ layers into consideration when generating branches at $i$-th layer is more reasonable and benefits the generation.

To incorporate the structural information of the previous layers into the generation of their subsequent layers, in this paper, we propose to encode the intermediate morphology which has been generated into an embedding and restrict the generation of branch pairs in the following layer to be conditioned on this embedding we obtain. Considering the conditional generation setting, we turn to use a seq2seq conditional variational autoencoder (CVAE) (Sohn et al., 2015) instead. We next present how we encode the structural information of the previous layers in detail.

We split the conditions extracted from the structural information of previous layers into two parts and name them **local** and **global** conditions respectively. Assuming that we are generating the pair highlighted in Fig. 2, we define the path from soma to the bifurcation from which the pair to be generated starts as the **local condition** and its previous layers structure as the **global condition**. We provide justifications for the significance of both local and global conditions from a neuron science perspective as follows and the details of encoding conditions are presented in Sec. 3.2.

**Justifications for the Conditions.** *Local:* Previous studies (Samsonovich & Ascoli, 2003; López-Cruz et al., 2011) show that the dendrites or axons usually extend away from the soma without making any sharp change of direction, thus reflecting that the orientation of a pair of sibling branches is mainly determined by the overall orientation of the path from the soma to the start point of the siblings. *Global:* Dendrites/axons establish territory coverage by following the organizing principle of self-avoidance (Sweeney et al., 2002; Sdrulla & Linden, 2006; Matthews et al., 2007). Self-avoidance refers to dendrites/axons that should avoid crossing, thus spreading evenly over a territory (Kramer & Kuwada, 1983). Since the global condition can be regarded as a set of the local conditions and each local condition can roughly decide the orientation of a corresponding pair of branches, the global condition helps us better organize the branches in the same layer and achieve an even spread.

**The Distinction of the Soma Branch Layer.** Under the aforementioned methodology formulation, we can observe that the soma branch layer differs from other layers in two folds. Firstly, 2 may not be a divisor of the branch number of the soma branch layer (i.e. $N^0 \mod 2 = 0$ may not hold). Secondly, the soma branch layer cannot be unified to the conditional generation formulation due to that there is no proper definition of conditions for it. Its distinction requires specific treatment.

In some other generation task settings (Liu et al., 2018; Liao et al., 2019), a little prior knowledge about the reference is introduced to the model as hints to enhance the realism of generations. Therefore, a straightforward solution is to adopt a similar approach: we can explicitly present the soma branches

as conditional input to the model, which are fairly small in number compared to all the branches[2]. Another slightly more complex approach is to generate the soma branch layer using another VAE without any conditions. In the experimental section, we demonstrated the results of both having the soma branch layer directly provided and not provided.

We have given a description of our generation procedure and will present the instantiations next.

## 3.2 MODEL INSTANTIATION

We use CVAE to model the distribution over branch pairs conditioned on the structure of their previous layers. The encoder encodes the branch pair as well as the condition to obtain a latent variable $\mathbf{z} \in \mathbb{R}^d$, where $d \in \mathbb{N}^+$ denotes the embedding size. The decoder generates the branch pairs from the latent variable $\mathbf{z}$ under the encoded conditions. Our goal is to obtain an encoder $f_\theta(\mathbf{z}|b_1, b_2, C)$ for a branch pair $(b_1, b_2)$ and condition $C$, and a decoder $g_\phi(b_1, b_2|\mathbf{z}, C)$ such that $g_\phi(f_\theta(b_1, b_2, C), C) \approx b_1, b_2$. The encoder and decoder are parameterized by $\theta$ and $\phi$ respectively.

**Encode Single Branch.** A branch $b_i$ can also be represented by a sequence of 3D coordinates, denoted as $b_i = \{v_{i_1}, v_{i_2}, \ldots, v_{i_{L(i)}}\}$, where $L(i)$ denotes the number of nodes included in $b_i$ and every pair of adjacent coordinates $(v_{i_k}, v_{i_{k+1}})$ is connected via an edge $e_{i_k, i_{k+1}}$. Notice that the length and the number of compartments constructing a branch vary from branch to branch, so as the start point $v_{i_1}$, making it difficult to encode the morphology information of a branch thereby. Thus we first translate the branch to make it start from the point $(0, 0, 0)$ in 3D space and then perform a resampling algorithm before encoding, which rebuilds the branches with $L$ new node coordinates and all compartments on a branch share the same length after the resampling operation. We present the details of the resampling algorithm in Appendix A. The branch $b_i$ after resampling can be written as $b_i' = \{v_{i_1}', v_{i_2}', \ldots, v_{i_L}'\}$. In the following text, all branches have been obtained after the resampling process. In principle, they should be denoted with an apostrophe. However, to maintain a more concise notation in the subsequent text, we will omit the use of the apostrophe.

Following MorphVAE (Laturnus & Berens, 2021), we use Long Short-Term Memory (LSTM) (Hochreiter & Schmidhuber, 1997) to encode the branch morphological information since a branch can be represented by a sequence of 3D coordinates. Each input coordinate will first be embedded into a high-dimension space via a linear layer, i.e. $\mathbf{x}_i = \mathbf{W}_{in} \cdot v_i'$ and $\mathbf{W}_{in} \in \mathbb{R}^{d \times 3}$. The obtained sequence of $\mathbf{x}_i$ is then fed into an LSTM. The internal hidden state $\mathbf{h}_i$ and the cell state $\mathbf{c}_i$ of the LSTM network for $i \geq 1$ are updated by:

$$\mathbf{h}_i, \mathbf{c}_i = \text{LSTM}(\mathbf{x}_i, \mathbf{h}_{i-1}, \mathbf{c}_{i-1}). \tag{1}$$

The initial two states $\mathbf{h}_0$ and $\mathbf{c}_0$ are both set to zero vectors. Finally, for a input branch $b$, we can obtain the corresponding representation $\mathbf{r}_b$ by concatenating $\mathbf{h}_L$ and $\mathbf{c}_L$, i.e.,

$$\mathbf{r}_b = \text{CONCAT}[\mathbf{h}_L, \mathbf{c}_L]. \tag{2}$$

**Encode Global Condition.** The global condition for a branch pair to be generated is obtained from the whole morphological structure of its previous layers. Note that the previous layers – exactly a subtree of the whole morphology, form an extremely sparse graph. Most nodes on the tree have no more than $3k$ $k$-hop[3] neighbors, thereby limiting the receptive field of nodes. Furthermore, as morphology is a hierarchical tree-like structure, vanilla Graph Neural Networks (Kipf & Welling, 2017) have difficulty in modeling such sparse hierarchical trees and encoding the global condition.

To tackle the challenging problem, instead of treating each coordinate as a node, we regard a branch as a node. Then the re-defined nodes will be connected by a directed edge $e_{b_i, b_j}$ starting from $b_i$ if $b_j$ is the succession of $b_i$. We can find the original subtree now is transformed into a forest $\mathcal{F} = (V^{\mathcal{F}}, E^{\mathcal{F}})$ composed of $N^0$ trees, where $N^0$ is the number of soma branches. $V^{\mathcal{F}}$ and $E^{\mathcal{F}}$ denote the set of nodes and edges in $\mathcal{F}$ respectively. The root of each included tree corresponds to a soma branch and each tree is denser than the tree made up of coordinate nodes before. Meanwhile, the feature of each re-defined node, extracted from a branch rather than a single coordinate, is far more informative than those of the original nodes, making the message passing among nodes more efficient.

Inspired by Tree structure-aware Graph Neural Network (T-GNN) (Qiao et al., 2020), we use a GNN combined with the Gated Recurrent Unit (GRU) (Chung et al., 2014) to integrate the hierarchical and

---

[2]For example, soma branches only account for approximately two percent of the total number in RGC dataset.
[3]Here, "$k$-hop neighbors of a node $v$" refers to all neighbors within a distance of $k$ from node $v$.

sequential neighborhood information on the tree structure to node representations. Similar to T-GNN, We also perform a bottom-up message passing within the tree until we update the feature of the root node. For a branch $b_i$ located at depth $k$ in the tree[4], its corresponding node feature $\mathbf{h}_{b_i}^{(k)}$ is given by:

$$\mathbf{h}_{b_i}^{(k)} = \begin{cases} \text{GRU}(\mathbf{r}_{b_i}, \sum_{b_j \in \mathcal{N}^+(b_i)} \mathbf{W}\mathbf{h}_{b_j}^{(k+1)}), & \mathcal{N}^+(b_i) \neq \emptyset \\ r_{b_i}, & \mathcal{N}^+(b_i) = \emptyset \end{cases}, \tag{3}$$

where $\mathbf{W}$ is a linear transformation, GRU is shared between different layers, and $\mathcal{N}^+(b_i)$ is the neighbors of $b_i$ who lies in deeper layer than $b_i$, which are exactly two subsequent branches of $b_i$. $\mathbf{r}_{bi}$ is obtained by encoding the branch using the aforementioned LSTM. The global feature is obtained by aggregating the features of root nodes be all trees in the forest $\mathcal{F}$:

$$\mathbf{h}_{global} = \text{READOUT} \left\{ \mathbf{h}_{b_i}^{(0)} | b_i \in R^{\mathcal{F}} \right\}, \tag{4}$$

where $R^{\mathcal{F}}$ denotes the roots of trees in $\mathcal{F}$ as and mean-pooling is used as the READOUT function.

**Encode Local Condition.** For a branch at the $l$-th layer, we denote the sequence of its ancestor branches as $\mathcal{A} = \{a_0, a_1, \ldots, a_{l-1}\}$ sorted by depth in ascending order where $a_0$ is a soma branch. As mentioned in Sec. 3.1, the growth of a branch is influenced by its ancestor branches. The closer ancestor branches might exert more influence on it. Thus we use Exponential Moving Average (EMA) (Lawrance & Lewis, 1977) to calculate the local feature. Denoting the feature aggregated from the first $k$ elements of $\mathcal{A}$ as $\mathbf{D}_k^{\mathcal{A}}$ and $\mathbf{D}_0^{\mathcal{A}} = \mathbf{r}_{a_0}$. For $k \geq 1$,

$$\mathbf{D}_k^{\mathcal{A}} = \alpha \mathbf{r}_{a_k} + (1 - \alpha)\mathbf{D}_{k-1}^{\mathcal{A}}, \quad 0 \leq \alpha \leq 1, \tag{5}$$

where $\alpha$ is a hyper-parameter. The local condition $\mathbf{h}_{local}$ for a branch at the $l$-th layer is $\mathbf{D}_{l-1}^{\mathcal{A}}$, i.e.,

$$\mathbf{h}_{local} = \mathbf{D}_{l-1}^{\mathcal{A}}. \tag{6}$$

The global condition ensures consistency within the layer, while the local condition varies among different pairs, only depending on their respective ancestor branches. Therefore, the generation order of branch pairs within the same layer does not have any impact, enabling synchronous generation.

**Design of Encoder and Decoder.** The encoder models a von-Mises Fisher (vMF) distribution (Xu & Durrett, 2018) with fixed variance $\kappa$ and mean $\mu$ obtained by aggregating branch information and conditions. The encoder takes a branch pair $(b_1, b_2)$ and the corresponding global condition $\mathcal{F}$, local condition $\mathcal{A}$ as input. All these input are encoded into features $\mathbf{r}_{b_1}$ $\mathbf{r}_{b_2}$, $\mathbf{h}_{global}$ and $\mathbf{h}_{local}$ respectively by the aforementioned procedure. The branches after resampling are denoted as $b_1' = \{v_1^{(1)}, v_2^{(1)}, \ldots, v_L^{(1)}\}$ and $b_2' = \{v_1^{(2)}, v_2^{(2)}, \ldots, v_L^{(2)}\}$. Then all the features are concatenated and fed to a linear layer $\mathbf{W}_{lat}$ to obtain the mean $\mu = \mathbf{W}_{lat} \cdot \text{CONCAT}[\mathbf{r}_{b_1}, \mathbf{r}_{b_2}, \mathbf{h}_{global}, \mathbf{h}_{local}]$. We take the average of five samples from $\mathbf{z}_i \sim \text{vMF}(\mu, \kappa)$ via rejection sampling as the latent variable $\mathbf{z}$.

As for the decoder $g_\phi(b_1, b_2 | \mathbf{z}, C = (\mathcal{A}, \mathcal{F}))$, we use another LSTM to generate two branches node by node from the latent variable $\mathbf{z}$ and condition representations, i.e. $\mathbf{h}_{global}$ and $\mathbf{h}_{local}$. First, we concatenate latent variable $z$ as well as the condition features $\mathbf{h}_{global}$ and $\mathbf{h}_{local}$ together and feed it to two different linear layers $\mathbf{W}_{sta1}, \mathbf{W}_{sta2}$ to determine the initial internal state for decoding branch $\hat{b}_1$ and $\hat{b}_2$. After that, a linear projection $\mathbf{W}'_{in} \in \mathbb{R}^{3 \times d}$ is used to project the first coordinates $v_1^{(1)}$ and $v_1^{(2)}$ of each branch into a high-dimension space $\mathbf{y}_1^{(1)} = \mathbf{W}'_{in} \cdot v_1^{(1)}, \mathbf{y}_1^{(2)} = \mathbf{W}'_{in} \cdot v_1^{(2)}$. Then the LSTM predicts $\mathbf{y}_{i+1}^{(t)}$ from $\mathbf{y}_i^{(t)}$ and its internal states repeatedly, where $t \in \{1, 2\}$. Finally, another linear transformation $\mathbf{W}_{out} \in \mathbb{R}^{3 \times d}$ is used to transform each $\mathbf{y}_i^{(t)}$ into coordinate $\hat{v}_i^{(t)} = \mathbf{W}_{out} \cdot \mathbf{y}_i^{(t)}, t \in \{1, 2\}$ and we can get the generated branch pair $\hat{b}_1 = \{\hat{v}_1^{(1)}, \hat{v}_2^{(1)}, \ldots, \hat{v}_L^{(1)}\}$ and $\hat{b}_2 = \{\hat{v}_1^{(2)}, \hat{v}_2^{(2)}, \ldots, \hat{v}_L^{(2)}\}$. In summary:

$$\mu = \mathbf{W}_{lat} \cdot \text{CONCAT}[\mathbf{r}_{b_1}, \mathbf{r}_{b_2}, \mathbf{h}_{global}, \mathbf{h}_{local}], \quad \mathbf{z}_i \sim \text{vMF}(\mu, \kappa), \quad \mathbf{z} = \frac{1}{5}\sum_{i=1}^{5} \mathbf{z}_i,$$

$$\mathbf{h}_0^{'(1)}, \mathbf{c}_0^{'(1)} = \mathbf{W}_{sta1} \cdot \text{CONCAT}[\mathbf{z}, \mathbf{h}_{global}, \mathbf{h}_{local}] \quad \mathbf{h}_0^{'(2)}, \mathbf{c}_0^{'(2)} = \mathbf{W}_{sta2} \cdot \text{CONCAT}[\mathbf{z}, \mathbf{h}_{global}, \mathbf{h}_{local}], \tag{7}$$

$$\mathbf{y}_1^{(t)} = \mathbf{W}'_{in} \cdot v_1^{(t)}, t \in \{1, 2\}, \quad \mathbf{h}_{i+1}^{'(t)}, \mathbf{c}_{i+1}^{'(t)} = \text{LSTM}(\mathbf{y}_i, \mathbf{h}_i^{'(t)}, \mathbf{c}_i^{'(t)}), t \in \{1, 2\},$$

$$\hat{v}_i^{(t)} = \mathbf{W}_{out} \cdot \mathbf{y}_i^{(t)}, t \in \{1, 2\}, \quad \hat{b}_t = \{\hat{v}_1^{(t)}, \hat{v}_2^{(t)}, \ldots, \hat{v}_L^{(t)}\}, t \in \{1, 2\}.$$

---

[4]Here, "tree" refers to the trees in $\mathcal{F}$ and not to the tree structure of the morphology.

Table 1: Performance on the four datasets by the six quantitative metrics. We leave MorphVAE's numbers on MBPL and MAPS blank because it may generate nodes with more than two subsequent branches that conflict with the definition of MBPL and MAPS for bifurcations. **MorphGrower** denotes the version where soma branches are directly provided. Meanwhile, **MorphGrower**[†] generates soma branches using another unconditional VAE. *Reference* corresponds to the statistical indicators derived from the realistic samples. A closer alignment with *Reference* indicates better performance. The best and the runner-up results are highlighted in **bold** and underline respectively.

| Dataset | Method | MBPL / μm | MMED / μm | MMPD / μm | MCTT / % | MASB / ° | MAPS / ° |
|---|---|---|---|---|---|---|---|
| VPM | *Reference* | $51.33 \pm 0.59$ | $162.99 \pm 2.25$ | $189.46 \pm 3.81$ | $0.936 \pm 0.001$ | $65.35 \pm 0.55$ | $36.04 \pm 0.38$ |
| | **MorphVAE** | $41.87 \pm 0.66$ | $126.73 \pm 2.54$ | $132.50 \pm 2.61$ | $0.987 \pm 0.001$ | —— | —— |
| | **MorphGrower** | $\mathbf{48.29 \pm 0.34}$ | $\mathbf{161.65 \pm 1.68}$ | $\mathbf{180.53 \pm 2.70}$ | $\underline{0.920 \pm 0.004}$ | $72.71 \pm 1.50$ | $\mathbf{43.80 \pm 0.98}$ |
| | **MorphGrower**[†] | $46.86 \pm 0.57$ | $159.62 \pm 3.19$ | $179.44 \pm 5.23$ | $\mathbf{0.929 \pm 0.006}$ | $59.15 \pm 3.25$ | $\underline{46.59 \pm 3.24}$ |
| RGC | *Reference* | $26.52 \pm 0.75$ | $308.85 \pm 8.12$ | $404.73 \pm 12.05$ | $0.937 \pm 0.003$ | $84.08 \pm 0.28$ | $50.60 \pm 0.13$ |
| | **MorphVAE** | $43.23 \pm 1.06$ | $248.62 \pm 9.05$ | $269.92 \pm 10.25$ | $0.984 \pm 0.004$ | —— | —— |
| | **MorphGrower** | $\mathbf{25.15 \pm 0.71}$ | $\mathbf{306.83 \pm 7.76}$ | $\mathbf{384.34 \pm 11.85}$ | $\mathbf{0.945 \pm 0.003}$ | $\mathbf{82.68 \pm 0.53}$ | $\mathbf{51.33 \pm 0.31}$ |
| | **MorphGrower**[†] | $23.32 \pm 0.52$ | $287.09 \pm 5.88$ | $358.31 \pm 8.54$ | $0.926 \pm 0.004$ | $76.27 \pm 0.86$ | $49.67 \pm 0.41$ |
| M1-EXC | *Reference* | $62.74 \pm 1.73$ | $414.39 \pm 6.16$ | $497.43 \pm 12.42$ | $0.891 \pm 0.004$ | $76.34 \pm 0.63$ | $46.74 \pm 0.85$ |
| | **MorphVAE** | $52.13 \pm 1.30$ | $195.49 \pm 9.91$ | $220.72 \pm 12.96$ | $0.955 \pm 0.005$ | —— | —— |
| | **MorphGrower** | $\mathbf{58.16 \pm 1.26}$ | $\mathbf{413.78 \pm 14.73}$ | $\mathbf{473.25 \pm 19.37}$ | $\underline{0.922 \pm 0.002}$ | $\mathbf{73.12 \pm 2.17}$ | $\mathbf{48.16 \pm 1.00}$ |
| | **MorphGrower**[†] | $54.63 \pm 1.07$ | $398.85 \pm 18.84$ | $463.24 \pm 22.61$ | $\mathbf{0.908 \pm 0.003}$ | $63.54 \pm 2.02$ | $48.77 \pm 0.87$ |
| M1-INH | *Reference* | $45.03 \pm 1.04$ | $396.73 \pm 15.89$ | $705.28 \pm 34.02$ | $0.877 \pm 0.002$ | $84.40 \pm 0.68$ | $55.23 \pm 0.78$ |
| | **MorphVAE** | $\underline{50.79 \pm 1.77}$ | $244.49 \pm 15.62$ | $306.99 \pm 23.19$ | $0.965 \pm 0.002$ | —— | —— |
| | **MorphGrower** | $\mathbf{41.50 \pm 1.02}$ | $\mathbf{389.06 \pm 13.54}$ | $\mathbf{659.38 \pm 30.05}$ | $\underline{0.898 \pm 0.002}$ | $\mathbf{82.43 \pm 1.41}$ | $\underline{61.44 \pm 4.23}$ |
| | **MorphGrower**[†] | $37.72 \pm 0.96$ | $349.66 \pm 11.40$ | $617.89 \pm 27.87$ | $\mathbf{0.876 \pm 0.002}$ | $78.66 \pm 1.12$ | $\mathbf{57.87 \pm 0.96}$ |

We jointly optimize the parameters $(\theta, \phi)$ of the encoder and decoder by maximizing the *Evidence Lower BOund* (ELBO), which is composed of a KL term and a reconstruction term:

$$\mathcal{L}(B = (b_1, b_2), C = (\mathcal{A}, \mathcal{F}); \theta, \phi) = \mathbb{E}_{f_\theta(\mathbf{z}|B,C)}[\log g_\phi(B|\mathbf{z}, C)] - \mathrm{KL}(f_\theta(\mathbf{z}|B,C) \| g_\phi(\mathbf{z}|C))], \quad (8)$$

where we abbreviate $(b_1, b_2)$ as $B$. We use a uniform vMF distribution $g_\phi(\mathbf{z}|C) = \mathrm{vMF}(\cdot, 0)$ as the prior. According to the property of vMF distribution, the KL term will become a constant depending on only the choice of $\kappa$. Then the loss function in Eq. 8 is reduced to the reconstruction term and can be rewritten as follows, estimated by the sum of mean-squared error between $(b_1, b_2)$ and $(\hat{b}_1, \hat{b}_2)$,

$$\mathcal{L}(B, C; \theta, \phi) = \mathbb{E}_{f_\theta(\mathbf{z}|B,C)}[\log g_\phi(B|\mathbf{z}, C)]. \quad (9)$$

### 3.3 SAMPLING NEW MORPHOLOGIES

Given a reference morphology $T$, a new morphology can be generated by calling our model regressively. In the generation of the $i$-th layer, the reference branch pairs on the $i$-th layer on $T$ and the generated previous $i - 1$ layers will be taken as the model input. We use $L_i = \{b_1^i, b_2^i, \ldots, b_{N^i}^i\}$ and $\hat{L}_i = \{\hat{b}_1^i, \hat{b}_2^i, \ldots, \hat{b}_{N^i}^i\}$ to represent the set of reference branches and those branches we generate at the $i$-th layer respectively. As mentioned in Sec. 3.1, we start the generation on the basis of the given soma branch layer. In general, the generation process can be formulated as:

$$\hat{T}^{(0)} = \{L_0\}, \quad \hat{T}^{(i)} = \hat{T}^{(i-1)} \cup \{\hat{L}_i\},$$
$$\hat{L}_i = g_\phi(f_\theta(L_i, \hat{T}^{(i-1)}), \hat{T}^{(i-1)}), \quad (10)$$

where $\hat{T}^{(i)}$ is the intermediate morphology after generating the $i$-th layer and $\hat{L}_i$ is the collection of generated branches at the $i$-th layer of the new morphology.

## 4 EXPERIMENTS

We evaluate the overall morphological statistics to directly compare the distribution of generation with the given reference in Sec. 4.2. Then we develop protocols to evaluate the generation plausibility and diversity in Sec. 4.3 and Sec. 4.4, respectively. Sec. 4.5 illustrates the dynamic snapshots of growing morphology which is a unique ability of our method compared to the peer work MorphVAE. We place some additional results, such as the ablation study, in Appendix J due to the limited space.

### 4.1 EVALUATION PROTOCOLS

**Datasets.** We use four popular datasets all sampled from adult mice: 1) **VPM** is a set of ventral posteromedial nucleus neuronal morphologies (Landisman & Connors, 2007; Peng et al., 2021); 2) **RGC** is a morphology dataset of retinal ganglion cell dendrites (Reinhard et al., 2019); 3) **M1-EXC**;

Figure 3: Distributions of four morphological metrics: MPD, BPL, MED and CTT on VPM dataset.

and 4) **M1-INH** refer to the excitatory pyramidal cell dendrites and inhibitory cell axons in M1. We split each dataset into training/validation/test sets by 8:1:1. See Appendix I for more dataset details.

**Baseline.** To our knowledge, there is no learning-based morphology generator except MorphVAE (Laturnus & Berens, 2021) which pioneers in generating neuronal morphologies resembling the given references. Grid search of learning rate over $\{1e-2, 5e-3, 1e-3\}$ and dropout rate over $\{0.1, 0.3, 0.5\}$ is performed to select the optimal hyper-parameters for MorphVAE. For fairness, both MorphVAE and ours use a 64 embedding size. Refer to Appendix H for more configuration details.

## 4.2 QUANTITATIVE RESULTS ON MORPHOLOGICAL STATISTICS

**Quantitative Metrics.** Rather than the Inception Score (IS) (Salimans et al., 2016) and Fréchet Inception Distance (FID) (Heusel et al., 2017) used in vision, there are tailored metrics as widely-used in computational neuronal morphology generation software tools. These metrics (as defined in L-measure (Scorcioni et al., 2008)) include: **BPL**, **MED**, **MPD**, **CTT**, **ASB**, **APS** which in fact are computed for a single instance. We further compute their mean over the dataset as **MBPL**, **MMED**, **MMPD**, **MCTT**, **MASB** and **MAPS**, and use these six metrics as our final metrics. The first four metrics relate to branch length and shape and the other two measure the amplitudes between branches. Detailed metric definitions are in Appendix B. Given the morphologies as references for generation from the dataset, we run MorphGrower/MorphVAE to generate the corresponding fake morphology.

As shown in Table 1, we can observe that the performance of our approach does not achieve a significant boost from re-using the soma branches as initial structures simply for convenience. Even without providing soma branches, MorphGrower still outperforms MorphVAE across all the datasets, except for the MBPL metric on the M1-INH dataset, suggesting the effectiveness of our progressive generation approach compared to the one-shot one in MorphVAE, especially for the four metrics related to branch length and shape. Due to the fact that MorphVAE uses 3D-walks as its basic generation units at a coarser granularity and resamples them, it loses many details of individual branches. In contrast, our MorphGrower uses branch pairs as its basic units at a finer granularity and performs branch-level resampling. A 3D-walk often consists of many branches, and with the same number of sampling points, our method can preserve more of the original morphology details. This is why MorphGrower performs better in the first four metrics. In addition, regarding MASB and MAPS that measure the amplitudes between branches and thus can reflect the orientations and layout of branches, the generation by MorphGrower is close to the reference data. We believe that this is owed to our incorporation of the neuronal growth principle, namely previous branches help in orientating and better organizing the subsequent branches. To further illustrate the performance gain, we plot the distributions of MPD, BPL, MED and CTT over the reference samples and generated morphologies on VPM in Fig. 3 (see the other datasets in Appendix N.2).

## 4.3 GENERATION PLAUSIBILITY WITH REAL/FAKE CLASSIFIER

The above six human-defined metrics can reflect the generation quality to some extent, while we seek a more data-driven way, especially considering the inherent complexity of the morphology data. Inspired by the scheme of generative adversarial networks (GAN) (Goodfellow et al., 2014), we here propose to train a binary classifier with the ratio $1:1$ for using the real-world morphologies as positive and generated ones as negative samples (one classifier for one generation method respectively). Then this classifier serves a role similar to the discriminator in GAN to evaluate the generation plausibility.

Specifically, features are prepared as input to the classifier. Given a 3-D neuron, we project its morphology onto the $xy$, $xz$ and $yz$ planes and obtain three corresponding projections. Inspired by the works in vision on multi-view representation (Prakash et al., 2021; Shao et al., 2022), we feed these three multi-view projections to three CNN blocks and obtain three corresponding representations. Here each CNN block adopts ResNet18 (He et al., 2016) as the backbone. Then, we feed each representation to a linear layer and obtain a final representation for each projection. Finally, the representations are concatenated and fed to the classifier, which is implemented with a linear layer followed by a sigmoid function. The whole pipeline is shown in Appendix H.3.

Table 2 reports the classification accuracy on the test set by the same data split as in Sec. 4.2. It is much more difficult for the neuronal morphologies generated by MorphGrower to be differentiated by the trained classifier, suggesting its plausibility. Moreover, we present the generated neuronal morphologies to neuroscience domain experts and receive positive feedback for their realistic looking.

## 4.4 GENERATION DIVERSITY EVALUATION

In neuronal morphology literature, **BlastNeuron Distance (BND)** (Wan et al., 2015) is a widely recognized metric for diversity, which measures the pairwise difference between two morphologies (see details in Appendix B.3). Due to its massive computing overhead for the whole dataset, here we use sub-sets of the datasets for evaluation. Specifically, we randomly select five neurons from the test set of each dataset as the reference morphologies and denote them as $T_1, T_2, ..., T_5$. For each $T_i$, we apply MorphVAE and MorphGrower to generate 50 corresponding samples, respectively. We randomly pick one from 50 generated samples and calculate its BND between other 49 samples.

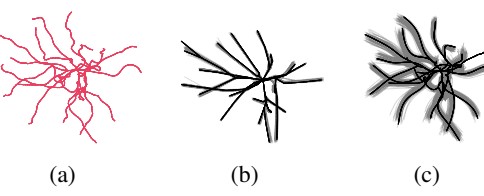

|     (a)      |      (b)      |      (c)      |

Figure 4: An example from VPM dataset. (a), (b) and (c) are the projections onto the $xy$ plane. (a): the reference; (b) and (c): samples generated by MorphVAE and MorphGrower, respectively.

Table 3 reports the average BND over these five randomly selected reference samples from the VPM dataset. Results show morphologies generated by MorphGrower consistently achieve higher average BNDs. In fact, the clustering operation in MorphVAE merges different 3D walks all together, which may reduce its diversity. Similar findings are observed in other datasets in Appendix J.4.

Fig. 4 visualizes the 2-D projection of the aforementioned 50 generated samples by MorphVAE and MorphGrower given the reference $T_1$. Specifically, in Fig. 4(b) and 4(c), we randomly pick a generated one and color it black, while the shading represents the other 49 samples. The samples generated by MorphGrower look more like the reference. Furthermore, we see that the width of shading in Fig. 4(c) is greater than that in Fig. 4(b), indicating the higher diversity.

Table 2: Classification accuracy (%). Accuracy approaching 50% indicates higher plausibility.

| Method \ Dataset | VPM | RGC | M1-EXC | M1-INH |
|---|---|---|---|---|
| **MorphVAE** | $86.75 \pm 06.87$ | $94.60 \pm 01.02$ | $80.72 \pm 10.58$ | $91.76 \pm 12.14$ |
| **MorphGrower** | $54.73 \pm 03.36$ | $62.70 \pm 05.24$ | $55.00 \pm 02.54$ | $54.74 \pm 01.63$ |

Table 3: Average BlastNeuron Distance on VPM.

| Method \ Reference | $T_1$ | $T_2$ | $T_3$ | $T_4$ | $T_5$ |
|---|---|---|---|---|---|
| **MorphVAE** | 3782.31 | 4616.16 | 1938.97 | 1713.91 | 6154.55 |
| **MorphGrower** | 4306.11 | 5635.86 | 6177.03 | 4905.25 | 11760.74 |

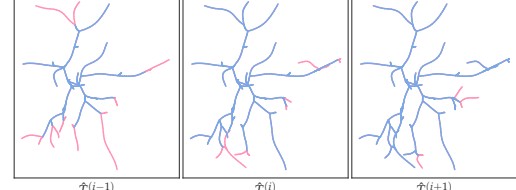

| $\hat{T}^{(i-1)}$ | $\hat{T}^{(i)}$ | $\hat{T}^{(i+1)}$ |

Figure 5: Projections onto $xy$ plane of three adjacent intermediate morphologies of a generated sample from RGC. For each $\hat{T}^{(j)}$, the pink represents newly generated layer $\hat{L}_j$ while the blue represents last intermediate morphology $\hat{T}^{(j-1)}$.

## 4.5 SNAPSHOTS OF THE GROWING MORPHOLOGIES

We present three adjacent snapshots of the intermediate generated morphologies of a randomly picked sample from the RGC in Fig. 5. More examples are given in Appendix N.3. We believe such studies are informative for the field, though have been rarely performed before.

## 5 CONCLUSION AND OUTLOOK

To achieve plausible and diverse neuronal morphology generation, we propose a biologically inspired neuronal morphology growing approach based on the given morphology as a reference. During the inference stage, the morphology is generated layer by layer in an auto-regressive manner whereby the domain knowledge about the growing constraints can be effectively introduced. Quantitative and qualitative results show the efficacy of our approach against the state-of-the-art baseline MorphVAE.

Our aspiration is that the fruits of our research will significantly augment the efficiency with which morphological samples of neurons are procured. This enhancement may pave the way for the generation of biologically realistic networks encompassing entire brain regions (Kanari et al., 2022), fostering a deeper understanding of the complex mechanisms that underpin neural computation.

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

# Appendix of MorphGrower

CONTENTS

## A   DATA PREPROCESS

### A.1   RESAMPLING ALGORITHM

The resampling algorithm is performed on a single branch and aims to solve the uneven sample node density over branches issue. $x \in (a, b)$ means that a coordinate $x$ is on the segment with endpoints $a$ and $b$ where $a$ and $b$ are also coordinates and we define the euclidean distance between $a$ and $b$ as $dist(a, b)$. Then for a branch $b = \{v_1, v_2, \ldots, v_l\}$, after resampling $b$, we can obtain $b' = \{v'_1, v'_2, \ldots, v'_{l'}\}$, where $v_i$ and $v'_i$ are 3D coordinates and $v'_i$ satisfies the following constraints:

$$
\begin{cases}
Len = \dfrac{\sum_{i=1}^{l-1} dist(v_i, v_{i+1})}{l' - 1}. \\
\forall\, j \in \{1, 2, \ldots, l'\}, \exists\, i \in \{1, 2, \ldots, l-1\},\; v'_j \in (v_i, v_{i+1}). \\
\forall\, v'_j \in (v_a, v_{a+1}) \text{ and } v'_{j+1} \in (v_b, v_{b+1}),\; \text{if } a \neq b,\; dist(v'_j, v_{a+1}) + \sum_{i=a+1}^{b-1} dist(v_i, v_{i+1}) + dist(v_b, v'_{j+1}) = Len. \\
\forall\, v'_j \in (v_a, v_{a+1}) \text{ and } v'_{j+1} \in (v_b, v_{b+1}),\; \text{if } a = b,\; dist(v'_j, v'_{j+1}) = Len.
\end{cases}
\tag{11}
$$

### A.2   PREPROCESSING OF THE RAW NEURONAL MORPHOLOGY SAMPLES

**Reason for preprocessing.**  Neuronal morphology is manually reconstructed from mesoscopic resolution 3D image stack, usually imaging from optical sectioning tomography imaging techniques to obtain images, and mechanical slicing yields a lower resolution in the z-axis than the resolution in xy-axis obtained from imaging. Limited by the resolution, the reconstruction algorithm can only connect two points by pixel-by-pixel, and the sampled point coordinates are aligned with the image coordinates, forming a series of jagged straight lines as shown in Fig. 6(a). Therefore, directly using the raw data for training is unsuitable and preprocessing is necessary.

To denoise the raw morphology samples, the preprocessing strategy we adopted in this paper is demonstrated as follows and it contains four key operations:

- **Merging pseudo somas.** In some morphology samples, there may exist multiple somas at the same time, where we call them pseudo somas. Due to that the volume of soma is relatively larger, we use multiple pseudo somas to describe the shape of soma and the coordinates of these pseudo somas are quite close to each other. In line with MorphoPy (Laturnus et al., 2020b), which is a python package tailored for neuronal morphology data, the coordinates of pseudo somas will be merged to the centroid of their convex hull as a final true soma.

- **Inserting Nodes.** As noted in Sec. 1, only the soma is allowed to have more than two outgoing branches (Bray, 1973; Wessells & Nuttall, 1978; Szebenyi et al., 1998). However, in some cases, two or even more bifurcations would be so close that they may share the same coordinate after reconstruction from images due to some manual errors. Notice that these cases are different from pseudo somas. We solve this problem by inserting points. For a non-soma node $v$ with more than two outgoing branches, we denote the set of the closest nodes on each subsequent branches as $\{v_{\text{close}_1}, v_{\text{close}_2}, \ldots, v_{\text{close}_{N_{\text{close}}}}\}$, where $N_{\text{close}}$ represents the number of branches extending away from $v$. We only reserve two edges $e_{v,v_{\text{close}_1}}$, $e_{v,v_{\text{close}_2}}$ and delete the rest edges $e_{v,v_{\text{close}_i}}$ for $3 \le i \le N_{\text{close}}$. Then we insert a new node which is located at the midpoint between node $v$ and $v_{\text{close}_1}$ and denote it as $v_{\text{mid}_1}$. Next we reconnect the $v_{\text{close}_3}, \ldots, v_{\text{close}_{N_{\text{close}}}}$ to node $v_{\text{mid}_1}$, i.e. adding edges $e_{v_{\text{mid}_1},v_{\text{close}_i}}$ for $3 \le i \le N_{\text{close}}$. We repeat such procedure until there is no other node having more than two outgoing edges apart from the soma.

- **Resampling branches.** We perform resampling algorithm described in Appendix A.1 over branches of each neuronal morphology sample. The goal of this step is two-fold: *i)* distribute the nodes on branches evenly so that the following smoothing step could be easier; *ii)* cut down the number of node coordinates to reduce IO workloads during training.

- **Smoothing branches.** As shown in Fig. 6(a), the raw morphology contains a lot of jagged straight line, posing an urgency for smoothing the branches. We smooth the branches via a sliding window. The window size is set to $2w + 1$, where $w$ is hyper-parameter. Given a branch $b_i = \{v_{i_1}, v_{i_2}, \ldots, v_{i_{L(i)}}\}$, for a node $v_{i_j}$ on $b_j$, if there are at least $w$ nodes on $b_i$ before it and at least $w$ nodes on $b_i$ after it, we directly calculate the average coordinate of $v_{i_j}$'s previous $w$ nodes, $v_{i_j}$'s subsequent $w$ nodes and $v_{i_j}$ itself. Then, we obtain a new node $v'_{i_j}$ at the average coordinate calculated. We denote the set of nodes that are involved in calculating the coordinate of $v'_{i_j}$ as $\mathcal{N}_{\text{smooth}}(v_{i_j})$. For those nodes which are close to the two ends of $b_i$, the number of nodes before or after them may be less than $w$. To solve this issue, we define $\mathcal{N}_{\text{smooth}}(v_{i_j})$ as follows:

$$\mathcal{N}_{\text{smooth}}(v_{i_j}) = \{v_{i_{j-p}}, v_{i_{j-p+1}}, \ldots, v_{i_{j-1}}, v_{i_j}, v_{i_{j+1}}, \ldots, v_{i_{j+s-1}}, v_{i_{j+s}}\}, \quad (12)$$

where $p = \min\{w, j-1, \eta(L(i) - j)\}$ and $s = \min\{w, L(i) - j, \eta(j-1)\}$. $\eta \in \mathbb{N}^+$ a is hyper-parameter in case of the extreme imbalance between the number of nodes before $v_{i_j}$ and the number of nodes after $v_{i_j}$. We reserve the two ends of $b_i$ and calculate new coordinates by averaging the coordinates of $\mathcal{N}_{\text{smooth}}(v_{i_j})$ for $2 \le j \le L(i) - 1$. Finally, we can obtain a smoother branch $b'_i = \{v_{i_1}, v'_{i_2}, \ldots, v'_{i_{L(i)-1}}, v_{i_{L(i)}}\}$.

**Remark.** Due to the low quality of raw **M1-EXC** and **M1-INH** datasets, the above data preprocessing procedure is applied to both of them. As for the raw **VPM** and **RGC** datasets, their quality is gooed enough. Therefore, we do not apply the above preprocessing procedure to them. We show an example morphology sample before and after preprocessing from M1-EXC dataset in Fig. 6.

## B    DEFINITIONS OF METRICS

In this section, we present the detailed definitions of all metrics we adopt for evaluation in this paper.

### B.1    METRIC ADOPTED IN SECTION 4.2

Before introducing the metrics, we first present the descriptions of their related quantitative characterizations as follows. Recall that all these adopted quantitative characterizations are defined on a single neuronal morphology.

**Branch Path Length (BPL).** Given a neuronal morphology sample $T$, we denote the set of branches it contains as $\mathcal{B}$. For a branch $b_i = \{v_{i_1}, v_{i_2}, \ldots, v_{i_{L(i)}}\}$, its path length is calculated

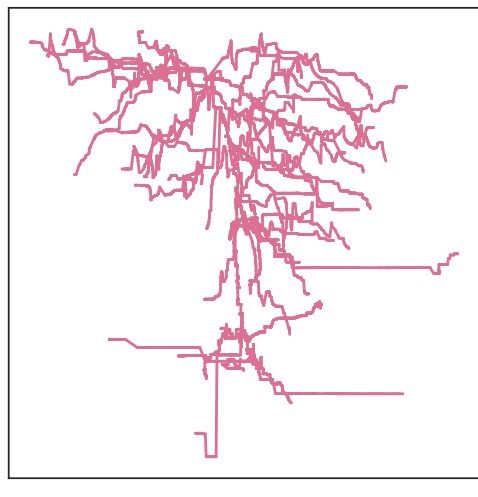 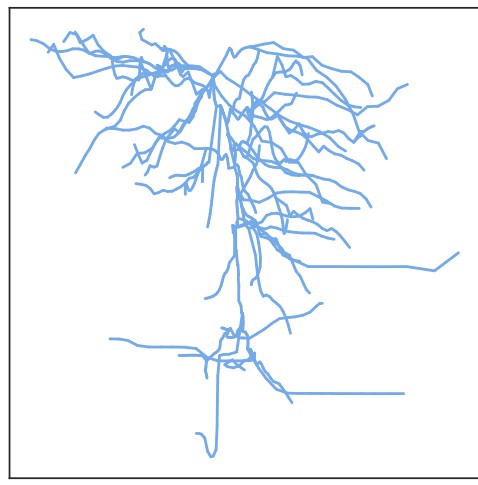

(a) Before preprocessing.          (b) After preprocessing.

Figure 6: An example morphology sample from M1-EXC dataset for illustrating the preprocessing. We demonstrate the projections of the morphology sample onto the $xy$ plane before and after preprocessing here.

by $\sum_{j=1}^{L(i)-1} dist(v_{i_j}, v_{i_{j+1}})$. BPL means the mean path length of branches over $T$, i.e.,

$$\text{BPL}(T) = \frac{1}{|\mathcal{B}|} \sum_{b_i \in \mathcal{B}} \sum_{j=1}^{L(i)-1} dist(v_{i_j}, v_{i_{j+1}}). \tag{13}$$

**Maximum Euclidean Distance (MED).** Given a neuronal morphology sample $T = (V, E)$, we denote the soma as $v_{soma}$. The $\text{MED}(T)$ is defined as:

$$\text{MED}(T) = \max_{v_i \in V} dist(v_i, v_{soma}). \tag{14}$$

**Maximum Path Distance (MPD).** Given a neuronal morphology sample $T = (V, E)$, we denote the soma as $v_{soma}$. For a node $v_i \in V$, $\{v_{soma}, v_{i^1}, v_{i^2}, \ldots, v_{i^{A_v(i)}}, v_i\}$ is the sequence of nodes contained on the path beginning from soma and ending at $v_i$, where $A_v(i)$ denotes the number of $v_i$'s ancestor nodes except soma. The path distance of $v_i$ is defined as $dist(v_{soma}, v_{i^1}) + \sum_{j=1}^{A_v(i)-1} dist(v_{i^j}, v_{i^{j+1}}) + dist(v_{i^{A_v(i)}}, v_i)$. Therefore, the maximum path distance (MPD) of $T$ is formulated as:

$$\text{MPD}(T) = \max_{v_i \in V} dist(v_{soma}, v_{i^1}) + \sum_{j=1}^{A_v(i)-1} dist(v_{i^j}, v_{i^{j+1}}) + dist(v_{i^{A_v(i)}}, v_i). \tag{15}$$

**ConTracTion (CTT).** CTT measures the mean degree of wrinkling of branches over a morphology $T$, and we denote the set of branches included in $T$ as $\mathcal{B}$. For a branch $b_i = \{v_{i_1}, v_{i_2}, \ldots, v_{i_{L(i)}}\}$, its degree of wrinkling is calculated by $\frac{dist(v_{i_1}, v_{i_{L(i)}})}{\sum_{j=1}^{L(i)-1} dist(v_{i_j}, v_{i_{j+1}})}$. Hence, the definition of CTT is:

$$\text{CTT}(T) = \frac{1}{|\mathcal{B}|} \sum_{b_i \in \mathcal{B}} \frac{dist(v_{i_1}, v_{i_{L(i)}})}{\sum_{j=1}^{L(i)-1} dist(v_{i_j}, v_{i_{j+1}})}. \tag{16}$$

**Amplitude between a pair of Sibling Branches (ASB).** Given a morphology sample $T$, we denote the set of bifurcations it contains as $V_{\text{bifurcation}}$. For any bifurcation node $v_i \in V_{\text{bifurcation}}$, there is a pair of sibling branches extending away from $v_i$. We denote the endpoints of these two branches as

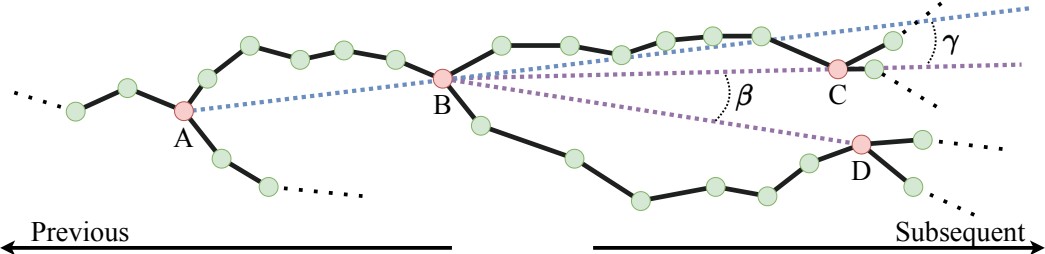

Figure 7: Illustrations for BAR and APS to facilitate understanding the deifinitions of them. Birfurcations are colored red.

$v_{i1}$ and $v_{i2}$, respectively. The order of sibling branches does not matter here. The vector $\overrightarrow{v_i v_{i1}}$ and $\overrightarrow{v_i v_{i2}}$ can form a amplitude like the amplitude $\beta$ in Fig. 7. ASB represents the mean amplitude size of all such instances over $T$. Hence, $\mathrm{ASB}(T)$ is defined as:

$$\mathrm{ASB}(T) = \frac{1}{|V_{\mathrm{bifurcation}}|} \sum_{v_i \in V_{\mathrm{bifurcation}}} \angle v_{i1} v_i v_{i2}. \tag{17}$$

**A**mplitude between of a **P**revious branch and its **S**ubsequent branch (**APS**). Given a morphology sample $T$, we denote the set of bifurcations it contains as $V_{\mathrm{bifurcation}}$. For any bifurcation node $v_i \in V_{\mathrm{bifurcation}}$, there is a pair of sibling branches extending away from $v_i$. Notice that $v_i$ itself is not only the start point of these but also the ending point of another branch. As demonstrated in Fig. 7, node $B$ is a bifurcation. There is a pair of sibling branches extending away from $B$. Node $C$ and $D$ are the ending points of the sibling branches. In the meanwhile, $B$ is the ending point of the branch whose start point is $A$. $\overrightarrow{AB}$ and $\overrightarrow{BC}$ can form a amplitude i.e. the amplitude $\gamma$. In general, we denote the start point of the branch ending with $v_i$ as $v_{i_{\mathrm{start}}}$. We use $v_{i_{\mathrm{end1}}}$ and $v_{i_{\mathrm{end2}}}$ to represent the ending points of two subsequent branches. $\mathrm{APS}(T)$ equals to the mean amplitude size of all amplitudes like $\gamma$ over $T$ and can be formulated as follows with $< \cdot, \cdot >$ representing the amplitude size between two vectors:

$$\mathrm{APS}(T) = \frac{1}{2 \cdot |V_{\mathrm{bifurcation}}|} \sum_{v_i \in V_{\mathrm{bifurcation}}} \angle < \overrightarrow{v_{i_{\mathrm{start}}} v_i}, \overrightarrow{v_i v_{i_{\mathrm{end1}}}} > + \angle < \overrightarrow{v_{i_{\mathrm{start}}} v_i}, \overrightarrow{v_i v_{i_{\mathrm{end2}}}} > . \tag{18}$$

Now, we have presented detailed definitions of all six quantitative characterizations. Recalling that the six metrics adopted are just the expectation of these six characterizations over the dataset, we denote the dataset as $\mathcal{T} = \{T_i\}_{i=1}^{N_T}$, where $N_T$ represents the number of morphology samples. **Then the six metrics are defined as below:**

- **Mean BPL (MBPL):**

$$\mathrm{MBPL} = \frac{1}{N_T} \sum_{i=1}^{N_T} \mathrm{BPL}(T_i). \tag{19}$$

- **Mean MED (MMED):**

$$\mathrm{MMED} = \frac{1}{N_T} \sum_{i=1}^{N_T} \mathrm{MED}(T_i). \tag{20}$$

- **Mean MPD (MMPD):**

$$\mathrm{MMPD} = \frac{1}{N_T} \sum_{i=1}^{N_T} \mathrm{MPD}(T_i). \tag{21}$$

- **Mean CTT (MCTT):**

$$\mathrm{MCTT} = \frac{1}{N_T} \sum_{i=1}^{N_T} \mathrm{CTT}(T_i). \tag{22}$$

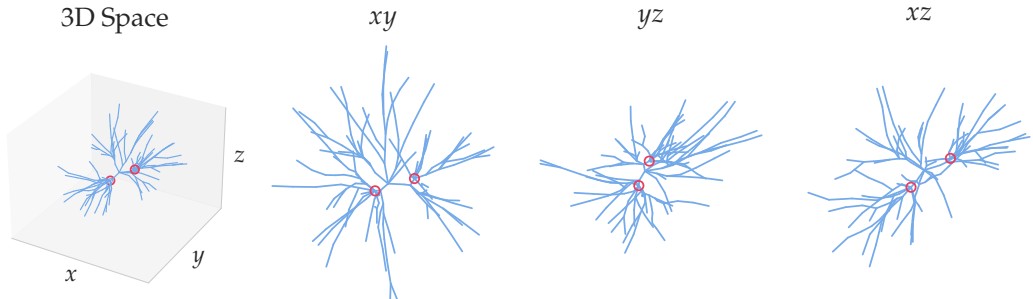

Figure 8: A topologically invalid neuronal morphology generated by MorphVAE. Nodes with more than two outgoing edges are circled in red. The first one demonstrates the morphology in 3D space. The three picture (from left to right) on the right are the projections of the sample onto $xy$, $yz$ and $xz$ planes, respectively.

- **Mean ASB (MASB):**

$$\text{MASB} = \frac{1}{N_T} \sum_{i=1}^{N_T} \text{ASB}(T_i). \tag{23}$$

- **Mean APS (MAPS):**

$$\text{MAPS} = \frac{1}{N_T} \sum_{i=1}^{N_T} \text{APS}(T_i). \tag{24}$$

### B.2 Metric Adopted in Section 4.3

In the discrimination test, we adopt the **Accuracy** as metric for evaluation, which is widely-used in classification tasks. We abbreviate the **true-positives**, **false-negatives**, **true-negatives** and **false-negatives** as **TP**, **FP**, **TN** and **FN**, respectively. Then Accuracy is formulated as:

$$\text{Accuracy} = \frac{|\text{TP}| + |\text{TN}|}{|\text{TP}| + |\text{TN}| + |\text{FP}| + |\text{FN}|}. \tag{25}$$

### B.3 Metric Adopted in Section 4.4

We use the average **BlastNeuron Distance (BND)** (Wan et al., 2015) among the generated samples of a given reference morphology $T$ to evaluate the diversity of generated samples. The BlastNeuron distance is defined on a pair of neuron morphologies and is based on topological structure and path shape alignment. For the given morphology pair, a series of morphological metrics are extracted from the neuronal morphologies to initially estimate the distance between the morphologies, and then an alignment approach based on graph matching combined with 3D coordinates is applied to identify local similarity. Since our numerous morphologies from the same sample are morphologically similar by nature, we only apply the local alignment algorithm to determine distances. We denote the BlastNeuron distance between two morphologies $T_1$ and $T_2$ as $\text{BND}(T_1, T_2)$. Higher $\text{BND}(T_1, T_2)$ indicates that the difference between $T_1$ and $T_2$ is greater. For a series of generated morphology $\{\hat{T}_1, \hat{T}_2, \ldots, \hat{T}_n\}$, the average BlastNeuron distance is defined as:

$$\frac{1}{n-1} \sum_{i=2}^{n} \text{BND}(\hat{T}_i, \hat{T}_1). \tag{26}$$

## C More Discussion on MorphVAE

Here we present examples to illustrate some limitations of MorphVAE (Laturnus & Berens, 2021).

Table 4: The topological validity rate of morphologies generated by MorphVAE on each dataset.

| Dataset | VPM | M1-EXC | M1-INH | RGC |
|---|---|---|---|---|
| validity rate | $0.052 \pm 0.019$ | $0.319 \pm 0.088$ | $0.038 \pm 0.025$ | $0.005 \pm 0.010$ |

## C.1 FAILURE TO ENSURE TOPOLOGICAL VALIDTY OF GENERATED MORPHOLOGIES

In the generation process, MorphVAE does not impose a constraint that only soma is allowed to have more than two outgoing edges, thereby failing to ensure the topological validty of generated samples. Here, we present an example generated by MorphVAE from VPM dataset, where there are more than one node that have more than two outgoing edges in Fig. 8.

**Elaboration on Topological Validity.**

- Under the majority of circumstances, neurons exhibit bifurcations (Lu & Werb, 2008; Peng et al., 2021), which signifies that branching points, excluding the soma node, typically have only two subsequent branches. Specifically, in the extensive morphological dataset presented in the work by Peng et al. (2021), the authors underscored the importance of treating trifurcations as topological errors that necessitate removal during post-processing quality assessment. Moreover, numerous tools for neuronal morphology analysis, such as the TREE toolbox (Cuntz et al., 2011), operate under the assumption that only binary neuron trees constitute valid and compatible data.

- In the infrequent instances where trifurcations have been observed, they were found to occur exclusively during the growth phase (Watanabe & Costantini, 2004). As a neuron reaches full maturation, these trifurcations often transform into two distinct bifurcations.

Consequently, there is no pragmatic rationale for intentionally generating or preserving trifurcations in a synthetic neuron. The indiscriminate and unregulated incorporation of trifurcations is ill-advised. Indeed, MorphVAE failed to adequately account for this factor, resulting in the generation of neurons with $3/4/5/n$-furcations.

We have computed the Topologically Validity Rate of samples generated by MorphVAE on the four datasets, indicating the proportion of samples that strictly adhere to a binary tree structure. The results are presented in Table 4, where we observe that the validity rate of MorphVAE is generally low. This highlights the significant improvement in topological plausibility offered by MorphGrower. Our generation mechanism ensures that our samples are all topologically valid.

Employing rejection sampling with the baseline MorphVAE to retain only topologically valid samples could be one potential approach. However, based on the results provided, especially on the RGC dataset where the validity rate is less than one percent, implementing rejection sampling might not yield a sufficient number of samples for a comprehensive evaluation.

## C.2 FAILURE TO GENERATE MORPHOLOGIES WITH RELATIVELY LONGER 3D-WALKS

Given a authentic morphology, MorphVAE aims to generate a resembling morphology. MorphVAE regards 3D-walks as the basic generation blocks. Note that the length of 3D-walks varies. MorphVAE determines to truncate those relatively longer 3D-walks during the encoding stage. Hence, in the final generated samples, its difficult to find morphologies with relatively longer 3D-walks. We provide two examples from M1-EXC dataset in Fig. 9 to show this limitation of MorphVAE.

## D SIGNIFICANT DISTINCTIONS FROM GRAPH GENERATION TASK

In this section, we aim to **highlight the significant differences between the neural morphology generation task discussed in this paper and the graph generation task.**

Graph generation primarily focuses on creating topological graphs with specific structural properties, whereas neuronal morphology generation aims to generate geometric shapes that conform to specific requirements and constraints within the field of neuroscience. Morphology is defined by a set of

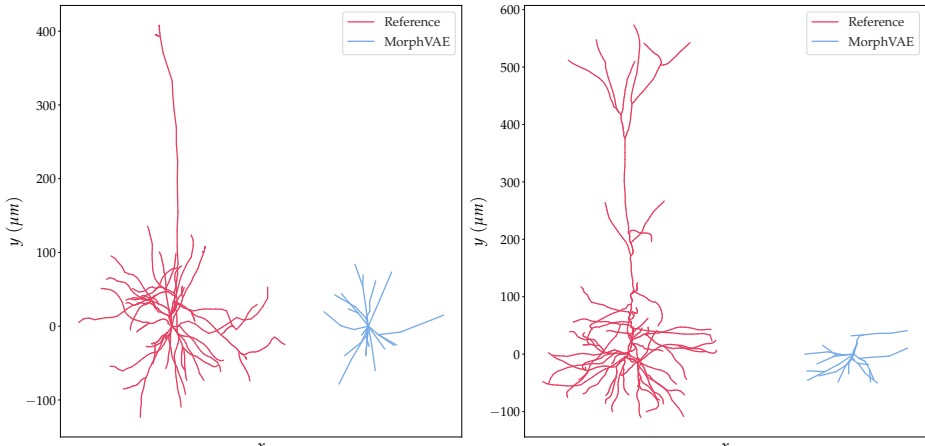

Figure 9: Each picture shows the projection of a pair of reference sample and the generated sample by MorphVAE, which is expected to resemble the target, onto the $xy$ plane. We can see that MorphVAE does fail to generate morphologies with relatively longer 3D-walks.

coordinate points in 3D space and their connection relationships. It is important to note that two different morphologies can share the same topological structure. For example, as illustrated in Fig. 10, the two neurons depicted in the first and second rows respectively possess the same topological structure in terms of algebraic topology (Hatcher, 2000) (i.e., the same Betti number-0 of 1 and the same Betti number-1 of 0), but are entirely distinct in morphology.

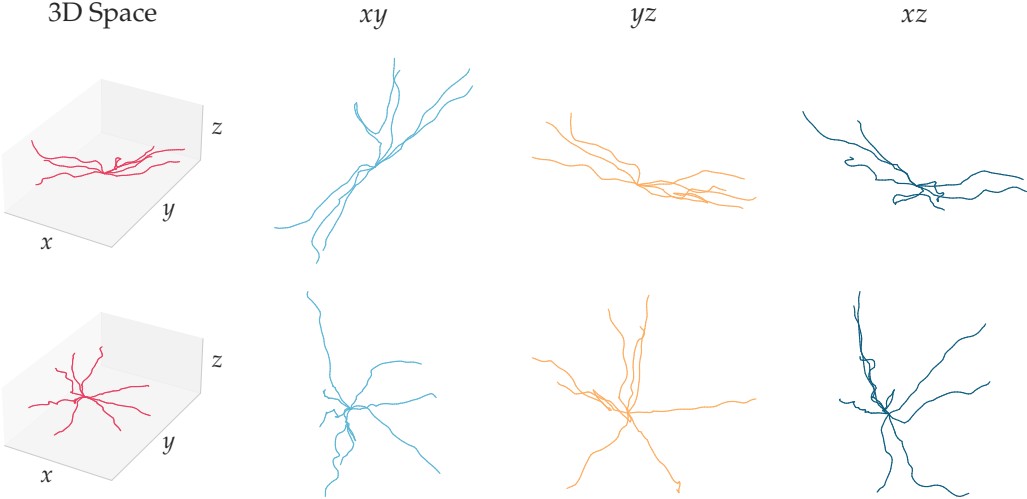

Figure 10: Two distinct neural morphology data are shown in two rows respectively. Both of them are trees composed of branches only connected by the root node (soma).

When attempting to adapt graph generation methods to the task of generating neuronal morphologies, scalability and efficiency pose significant challenges. Neuronal morphologies typically have far more nodes than the graphs used for training in graph generation. For example, the average number of nodes in the QM9 dataset (Ruddigkeit et al., 2012; Ramakrishnan et al., 2014), which is a commonly used dataset for molecular generation tasks, is **18**, while the average number of nodes in the VPM dataset (Landisman & Connors, 2007; Peng et al., 2021), the dataset used in our paper, is **948**. Graph generation methods require generating an adjacency matrix with $\Theta(n^2)$ size, where $n$ is the number of nodes, resulting in unaffordable computational overhead.

The most significant difference between graph and morphology generation is that every neuron morphology is a tree with special structural constraints. Thus, the absence of loops is a crucial criterion for valid generated morphology samples. We surveyed numerous graph generation-related articles and found that existing algorithms do not impose explicit constraints to ensure loop-free graph samples. This is primarily due to the application scenarios of current graph generation tasks, which mainly focus on molecular generation. Since molecules naturally have the possibility of containing loops, there is no need to consider the no-loop constraint. Furthermore, as stated in Appendix C.1, in most cases, neurons have only bifurcations (Lu & Werb, 2008; Peng et al., 2021) (i.e., branching points except for the soma node have only two subsequent branches). This implies that the trees in this study, except for the soma node, must be at most binary, resulting in even more stringent generated constraints. Nevertheless, current graph generation methods cannot ensure strict compliance with the aforementioned constraints.

Overall, there are significant differences between the graph generation task and the neuronal morphology generation task, which make it non-trivial to adapt existing graph generation methods to the latter. This is also why neither MorphVAE nor our paper chose any existing graph generation method as a baseline for comparison.

# E    SUPPLEMENTARY CLARIFICATIONS FOR RESAMPLING OPERATION

Although recurrent neural networks (RNNs) are able to encode or generate branches of any length, the second resampling operation is still necessary for the following reasons. Unlike numerous sequence-to-sequence tasks in the natural language processing (NLP) field, neuron branches are composed of 3D coordinates, which are continuous data type rather than discrete type. Therefore, it is difficult to use an approach like outputting an <END> token to halt generation. Additionally, it is challenging to calculate reconstruction loss between branches of different lengths.

In our experiments, we resample each branch to a fixed number of nodes, setting the hyperparameter at 32, which exceeds the original node count for most branches in the raw data. To demonstrate that our resampling operation does not have a significant impact on the morphology of neurons, we conducted the following statistics. We refer to branches with over 32 nodes in the raw data as "large branches". As shown in Table 5, for the majority of branches, we perform upsampling rather than downsampling, thereby preserving the morphology well.

Table 5: Statistics related to large branches.

| Dataset | VPM | M1-EXC | M1-INH | RGC |
|---|---|---|---|---|
| total # of branches | 26,395 | 18,088 | 114,139 | 232,315 |
| total # of large branches | 2,127 | 0 | 0 | 92 |
| percentage $\frac{\text{large}}{\text{all}}$ / % | 8.05 | 0.00 | 0.00 | 0.00 |

We designed an experiment to further investigate the impact of our resampling operation on the original branches, particularly on the large branches. We defined a metric called path length difference (abbreviated as PLD) to reflect the difference between two branches. We use PLD to measure the difference between the same branch before and after resampling. For two branches $b_1 = \{x^{(1)}, x^{(2)}, \cdots, x^{(i)}\}$ and $b_2 = \{y^{(1)}, y^{(2)}, \cdots, y^{(j)}\}$ the PLD is defined as:

$$\left| \sum_{c=1}^{i-1} \sqrt{\sum_{d=1}^{3}(x_d^{(c)} - x_d^{(c+1)})^2} - \sum_{c=1}^{j-1} \sqrt{\sum_{d=1}^{3}(y_d^{(c)} - y_d^{(c+1)})^2} \right|. \tag{27}$$

The result is shown in Table 6 and since there are no large branches in M1-EXC and M1-INH, the corresponding results are missing.

From the table above, we can observe that compared to the original branch lengths, the length differences of the branches after resampling are quite small across all datasets, even for large branches. This further indicates that our resampling operation can effectively preserve the original neuronal morphology.

Table 6: PLD results for branches before and after resampling. Due to the absence of large branches in the M1-EXC and M1-INH datasets, data related to large branches are missing on the two datasets.

| Dataset | RGC | VPM | M1-EXC | M1-INH |
|---|---|---|---|---|
| avg. PLD on all branches | $0.13 \pm 0.42$ | $0.17 \pm 0.26$ | $0.37 \pm 0.68$ | $0.30 \pm 0.70$ |
| avg. path length of all original branches | $24.31 \pm 25.30$ | $52.20 \pm 46.29$ | $60.32 \pm 51.03$ | $43.30 \pm 43.48$ |
| avg. PLD on large branches | $7.65 \pm 3.16$ | $0.65 \pm 0.37$ | —— | —— |
| avg. path length of all large branches | $264.29 \pm 44.00$ | $139.82 \pm 14.02$ | —— | —— |

## F  MOTIVATION OF THE CHOICE OF ARCHITECTURES

In the main text, we have already presented the rationale behind our selection of specific architectures. Here, we provide a concise summary of the motivation behind our choice of key architectures.

### F.1  USING LSTM AS OUR BRANCH ENCODER

Our baseline MorphVAE incorporates LSTM as its branch encoder and decoder. To ensure a fair comparison, we have followed the same approach as our baseline and also employed LSTM as our branch encoder.

Due to our limited data samples, we are uncertain if more complex sequence-based models like the Transformer would be effectively trainable. As an experiment, we replace our LSTM branch encoder with a Transformer, and the results, which are presented in Appendix J.1, indicate that the Transformer does not demonstrate significant advantages over the LSTM in terms of performance. This suggests that the LSTM is already sufficiently powerful for this task.

### F.2  USING vMF-VAE

The vMF-VAE with a fixed $\kappa$ is proposed to prevent the KL collapse typically observed in the Gaussian VAE setting (Xu & Durrett, 2018; Davidson et al., 2018). By fixing $\kappa$, the KL distance between the posterior distribution $\text{vMF}(\cdot, \kappa)$ and the prior distribution $\text{vMF}(\cdot, 0)$ remains constant, thus avoiding the KL collapse.

### F.3  TREATING A BRANCH AS A NODE AND EMPLOYING T-GNN-LIKE NETWORKS.

The reasons for this aspect have already been explained in great detail in Section 3.2 *Encode Global Condition* of the main text, so we will not reiterate them here.

## G  LIMITATIONS AND FUTURE WORK

### G.1  LIMITATIONS

There are two aspects limiting our approach, both in terms of data availability. First, we only have morphology information regarding the coordinates of the nodes, hence the generation may inherently suffer from the illness of missing information (e.g. the diameter). Second, the real-world samples for learning do not reflect the dynamic growing procedure and thus learning such dynamics can be challenging. In our paper, we have simplified the procedure by imposing synchronized layer growth which in reality could be asynchronous to some extent with growing randomness. In both cases, certain prior knowledge need to be introduced as preliminarily done in this paper which could be further improved for future work.

### G.2  FUTURE DIRECTIONS

#### G.2.1  INCLUSION OF THE BRANCH DIAMETER

In our paper, we focus on the three-dimensional spatial coordinates of each node, with each node corresponding to a three-dimensional representation. To incorporate the diameter, we would simply

add an extra dimension to the node representation, expanding the original three-dimensional representation to four dimensions. Incorporating the diameter into our method is relatively straightforward and does not necessitate significant modifications to our model design. Furthermore, diameter variations within neurons are generally smooth. Even when significant changes in diameter occur, as noted in the work by Conde-Sousa et al. (2017), we generally ascribe these inconsistencies to problems encountered during tissue preprocessing and the staining procedure. This indicates that differences in this dimension are less significant compared to the other three dimensions (spatial coordinates), making the training process for this dimension more stable and easier for the model to learn. Initially, we considered including the diameter in our model learning process, but given that the neuron's 3D geometry is of greater importance than the diameter and remains the primary focus, we ultimately chose not to include it. Additionally, due to limitations in imaging technology, not all neuron datasets contain diameter information, which is another reason why we did not take diameter into account. However, it is important to note that diameter information is essential for characterizing more specific neurons. We believe that as neuronal imaging technology advances, this information will gradually become more available. With sufficient training data and the inclusion of diameter information, our method should be capable of generating more detailed neurons.

### G.2.2 Inclusion of the Spines or Boutons

As mentioned in the limitations section of our paper, the existing data only contains spatial coordinate information, lacking additional input. If we could obtain splines/boutons information, we have a simple and feasible idea: for each spline/bouton, we can learn a feature and then share that feature with the surrounding nodes. This can be concatenated with the features they already encode, forming a new representation for each node. This design can make use of the additional splines/boutons information provided and does not require significant modifications to our model. However, this idea is just an initial simple attempt, and there may be better ways to utilize this extra information. Similarly, to the diameter mentioned earlier, data on spines/boutons is also scarce at this stage. If we have access to a wealth of such data, we believe it could further improve our neuron morphology generation results.

### G.2.3 Use of Dense Imaging Data

We believe that the dense imaging data can serve two main purposes as below:

- Since dense imaging data encompasses a vast amount of information about individual neurons, we are confident that it will facilitate the enhanced generation of single neuron morphologies in the future.

- Dense imaging data illustrates the intricate connections between neurons, which will prove highly valuable for generating simulated neuron populations down the line.

### G.3 Other Potential Applications

Our approach is not limited to the neuronal morphology generation task alone. In fact, our method can be adapted to generate data with similar tree-like branching structures in morphology (or even beyond). For instance, retinal capillaries are also scarce in data, and we can attempt to generate more retinal capillary data using our method to address this data insufficiency issue. With additional capillary samples, the related segmentation models can be trained more effectively, promoting the development of biomedical engineering and bioimaging fields. Therefore, the scope of our method's application is not narrow.

Moreover, as emphasized in the introduction section of our paper, the current process of obtaining single neuron morphology data is time-consuming and labor-intensive. Therefore, the motivation behind our method is to design an efficient generation method for single neuron morphology samples. In the future, we can also try to use our method as a basic building block to construct a neuron population, which will help us gain a deeper understanding of information transmission within the nervous system and further our knowledge of brain function.

# H    IMPLEMENTATION DETAILS

This section describes the detailed experimental setup or training configurations for MorphVAE and MorphGrower in order for reproducibility. We firstly list the configurations of our environments and packages as below:

- Ubuntu: 20.04.2

- CUDA: 10.1

- Python: 3.7.0

- Numpy: 1.21.6

- Pytorch: 1.12.1

- PyTorch Geometric: 2.1.0

- Scipy: 1.7.3

Experiments are repeated 5 times with mean and standard deviation reported, running on a machine with i9-10920X CPU, RTX 3090 GPU and 128G RAM.

## H.1    MORPHVAE

**Resampling Distance.** MorphVAE (Laturnus & Berens, 2021) regards a 3D-walk as a basic building block when generating neuronal morphologies. Following the data preparation steps of MorphVAE, all the morphologies are first resampled at a certain distance and then scaled down according to the resampling distance to make the training and clustering easier. The resampling distances over datasets are all shown in Table 7.

Table 7: The resample distance for each dataset. We adopt the default distance used in the original paper for RGC, M1-EXC and M1-INH. As for VPM, which is not adopted in MorphVAE for evaluation, we set its corresponding resampling distance to 30 μm.

| Dataset | VPM | RGC | M1-EXC | M1-INH |
|---|---|---|---|---|
| **Resampling Distance / μm** | 30 | 30 | 50 | 40 |

**Hyper Parameter Selection.** Apart from the hyper-parameters mentioned in Section 4, we also perform grid search for the supervised-learning proportion over $\{1, 0.5, 0.1, 0\}$ on all datasets except **VPM** because no cell type label is provided for **VPM** and the classifier is not set up for this dataset. Following the original paper, we set the 3D walk length as 32 and $\kappa = 500$. The teaching-force for training decoder is set to $0.5$. As for the distance threshold for agglomerative clustering, we use $0.5$ for **VPM**, **M1-EXC**, **M1-INH** and $1.0$ for **RGC**.

**Pretraining.** Following the original paper of MorphVAE, to better encode the branch information, we pretrain MorphVAE on the artificially generated toy dataset. Then model weight of pretrained LSTM is also used to initialize the LSTM part of the single branch encoder in our MorphGrower.

## H.2    OUR METHOD

We implement our method in Pytorch and adopt grid search to tune the hyper parameters. The learning rate is searched within $\{5e - 2, 1e - 2, 5e - 3, 1e - 3, 5e - 4, 1e - 4, 5e - 5, 1e - 5\}$ and dropout rate is searched within $\{0.1, 0.3, 0.5, 0.7\}$. In line with MorphVAE, we also the set the teaching-force for training decoder to $0.5$. In line with morphvae, we use $\kappa = 500$ for training. The number of nodes rebuilding a branch, i.e. the hyper parameter $L$, is set to 32. The weight $\alpha$ in Eq. 5 is set to $0.5$. Besides, the hyper-parameter $m$ in Eq. 4 is set to 2. We adopt Adam optimizer to optimize learnable parameters.

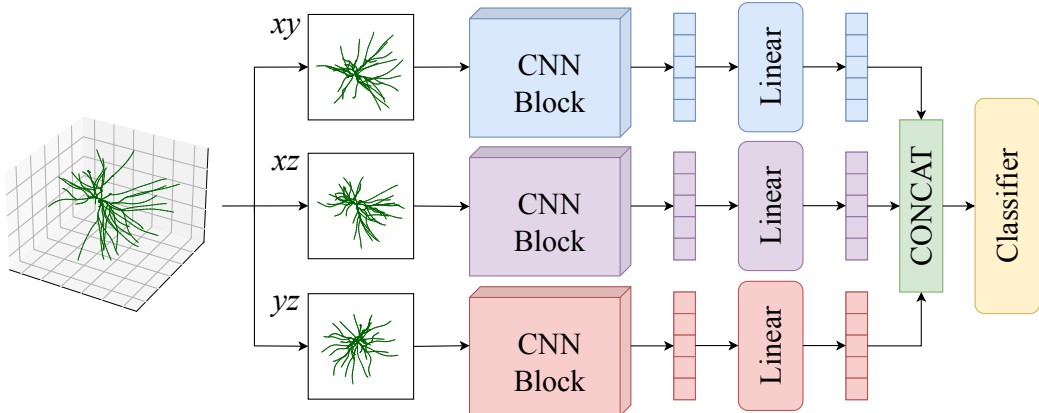

Figure 11: The overview of the pipeline for the discrimination test.

## H.3 SUPPLEMENTARY DESCRIPTIONS FOR SECTION 4.3

**Rationale behind evaluating plausibility using the CV method.** Neurons have complicated, unique and type-defining structures, making their visual appearances a discriminative characteristics. It is important that a synthetic neuron "looks" real, either by human or by a computer. Thus, as an additional proof, we designed a vision-based classifier and tried to see if the generated neurons can "deceive" it. As reported in Table 2, the classifier cannot well differentiate our generated data from the real ones, while the MorphVAE results can be well identified. Furthermore, our use of three views as inputs to assess the plausibility of the generated neuronal morphology samples is well-founded. This can be supported by many cell typing studies (Jefferis et al., 2007; Sümbül et al., 2014; Laturnus et al., 2020a), researchers adopt density maps as descriptors for samples, where the so-called density map is essentially the projection of a sample's 3D morphology onto a specific coordinate plane.

**Learning Objective.** We adopt the **Focal-Loss** as the learning objective for training the classifier, which is widely-used in classification tasks. It is a dynamically scaled Cross-Entropy-Loss, where the scaling factor decays to zero as confidence in the correct class increases (Lin et al., 2017).

**Pipeline Overview.** We demonstrate the whole pipeline in Fig. 11.

**Hyper Parameter Selection.** We search the optimal hyper-parameters by ranging learning rate over $\{1e-3, 5e-3, 1e-4, 5e-4, 1e-5, 5e-5\}$ and dropout rate over $\{0.2, 0.5, 0.8\}$. Adam optimizer is adopted to optimize learnable parameters.

## H.4 SUPPLEMENTARY DESCRIPTIONS FOR SECTION 4.4

The BlastNeuron Distance, which has been introduced in Appendix B.3, is relatively sensitive to the number of coordinates on the given morphologies. For fair comparison, we resample all the generated samples obtain from MorphVAE and MorphGrower at a distance of 2 μm, using the python package MorphoPy (Laturnus et al., 2020b).

## I MORE DATASET INFORMATION

The statistics of the four datasets are reported in Table 8 and their download links are as follows:

- **VPM**: https://download.brainimagelibrary.org/biccn/zeng/luo/fMOST/
- **RGC**: https://osf.io/b4qtr/wiki/home/
- **M1-EXC & M1-INH**: https://github.com/berenslab/mini-atlas

**Remark.** Axons in the VPM dataset typically have long-range projections, which can be dozens of times longer than other branches, and they target specific brain regions that they must innervate.

Table 8: **Datasets Statistics.** "#train/#valid/#test" denotes the number of samples in the training/validation/test set, respectively. "avg. / max # of branches" represents the average and maximum number of branches that a neuron contains respectively.

| Dataset | #train | #valid | #test | avg. / max # of branches |
|---------|--------|--------|-------|--------------------------|
| **VPM**     | 266 | 57  | 57  | 69.46/153    |
| **RGC**     | 534 | 114 | 115 | 303.47/2327  |
| **M1-EXC**  | 192 | 41  | 42  | 65.77/156    |
| **M1-INH**  | 259 | 55  | 57  | 307.65/913   |

Table 9: The result of ablation study on the four datasets by six quantitative metrics. The best and the runner-up in each columns are highlighted in **bold** and underline respectively. We leave MorphVAE's numbers on MBPL and MAPS blank because it may generate nodes with more than two subsequent branches that conflict with the definition of MBPL and MAPS for bifurcations. A closer alignment with *Reference* indicates better performance.

| Dataset | Method | MBPL / μm | MMED / μm | MMPD / μm | MCTT / % | MASB / ° | MAPS / ° |
|---------|--------|-----------|-----------|-----------|----------|----------|----------|
| **VPM** | *Reference* | $51.33 \pm 0.59$ | $162.99 \pm 2.25$ | $189.46 \pm 3.81$ | $0.936 \pm 0.001$ | $65.35 \pm 0.55$ | $36.04 \pm 0.38$ |
| | **MorphVAE** | $41.87 \pm 0.66$ | $126.73 \pm 2.54$ | $132.50 \pm 2.61$ | $0.987 \pm 0.001$ | —— | —— |
| | **MorphGrower** | $\mathbf{48.29 \pm 0.34}$ | $161.65 \pm 1.68$ | $180.53 \pm 2.70$ | $0.920 \pm 0.004$ | $72.71 \pm 1.50$ | $\mathbf{43.80 \pm 0.98}$ |
| | LSTM → Transformers | $47.40 \pm 0.88$ | $\mathbf{162.46 \pm 3.82}$ | $\mathbf{180.87 \pm 3.09}$ | $0.943 \pm 0.010$ | $70.94 \pm 2.77$ | $53.86 \pm 0.99$ |
| | - Local Condition | $40.90 \pm 0.79$ | $137.47 \pm 2.63$ | $162.53 \pm 3.24$ | $0.911 \pm 0.006$ | $74.55 \pm 0.88$ | $58.22 \pm 0.32$ |
| | - Global Condition | $44.51 \pm 0.78$ | $153.84 \pm 3.56$ | $173.58 \pm 5.41$ | $\underline{0.938 \pm 0.003}$ | $71.29 \pm 3.56$ | $47.06 \pm 0.59$ |
| | - EMA | $45.24 \pm 0.22$ | $155.11 \pm 1.98$ | $173.68 \pm 2.86$ | $\mathbf{0.936 \pm 0.005}$ | $67.79 \pm 1.45$ | $\underline{46.28 \pm 0.08}$ |
| **RGC** | *Reference* | $26.52 \pm 0.75$ | $308.85 \pm 8.12$ | $404.73 \pm 12.05$ | $0.937 \pm 0.003$ | $84.08 \pm 0.28$ | $50.60 \pm 0.13$ |
| | **MorphVAE** | $43.23 \pm 1.06$ | $248.62 \pm 9.05$ | $269.92 \pm 10.25$ | $0.984 \pm 0.004$ | —— | —— |
| | **MorphGrower** | $\mathbf{25.15 \pm 0.71}$ | $\underline{306.83 \pm 7.76}$ | $\underline{384.34 \pm 11.85}$ | $\mathbf{0.945 \pm 0.003}$ | $82.68 \pm 0.53$ | $\underline{51.33 \pm 0.31}$ |
| | LSTM → Transformers | $\underline{25.10 \pm 0.65}$ | $\mathbf{308.35 \pm 7.34}$ | $\mathbf{387.67 \pm 10.55}$ | $\underline{0.948 \pm 0.003}$ | $\mathbf{84.04 \pm 0.33}$ | $52.35 \pm 0.14$ |
| | - Local Condition | $23.56 \pm 0.74$ | $294.01 \pm 8.21$ | $363.86 \pm 11.36$ | $0.954 \pm 0.003$ | $79.67 \pm 1.17$ | $54.44 \pm 0.36$ |
| | - Global Condition | $22.99 \pm 0.83$ | $293.87 \pm 9.01$ | $354.95 \pm 11.85$ | $0.954 \pm 0.006$ | $78.19 \pm 4.10$ | $\mathbf{50.96 \pm 0.63}$ |
| | - EMA | $23.38 \pm 0.66$ | $295.09 \pm 8.76$ | $359.76 \pm 8.76$ | $0.951 \pm 0.005$ | $78.47 \pm 1.84$ | $52.25 \pm 0.44$ |
| **M1-EXC** | *Reference* | $62.74 \pm 1.73$ | $414.39 \pm 6.16$ | $497.43 \pm 12.42$ | $0.891 \pm 0.004$ | $76.34 \pm 0.63$ | $46.74 \pm 0.85$ |
| | **MorphVAE** | $52.13 \pm 1.30$ | $195.49 \pm 9.91$ | $220.72 \pm 12.96$ | $0.955 \pm 0.005$ | —— | —— |
| | **MorphGrower** | $\mathbf{58.16 \pm 1.26}$ | $\mathbf{413.78 \pm 14.73}$ | $\mathbf{473.25 \pm 19.37}$ | $\mathbf{0.922 \pm 0.002}$ | $73.12 \pm 2.17$ | $\mathbf{48.16 \pm 1.00}$ |
| | LSTM → Transformers | $\underline{56.75 \pm 1.49}$ | $415.90 \pm 4.39$ | $\underline{472.30 \pm 7.99}$ | $0.942 \pm 0.005$ | $72.97 \pm 1.75$ | $51.06 \pm 0.98$ |
| | - Local Condition | $55.85 \pm 1.24$ | $409.66 \pm 7.36$ | $464.18 \pm 9.07$ | $0.940 \pm 0.004$ | $\mathbf{73.81 \pm 1.24}$ | $51.54 \pm 0.84$ |
| | - Global Condition | $55.01 \pm 0.65$ | $404.42 \pm 8.67$ | $453.58 \pm 11.90$ | $0.955 \pm 0.007$ | $71.72 \pm 1.23$ | $\underline{48.48 \pm 1.04}$ |
| | - EMA | $55.71 \pm 1.24$ | $407.29 \pm 16.28$ | $458.49 \pm 12.29$ | $0.951 \pm 0.008$ | $72.61 \pm 4.35$ | $50.20 \pm 0.83$ |
| **M1-INH** | *Reference* | $45.03 \pm 1.04$ | $396.73 \pm 15.89$ | $705.28 \pm 34.02$ | $0.877 \pm 0.002$ | $84.40 \pm 0.68$ | $55.23 \pm 0.78$ |
| | **MorphVAE** | $\underline{50.79 \pm 1.77}$ | $244.49 \pm 15.62$ | $306.99 \pm 23.19$ | $0.965 \pm 0.002$ | —— | —— |
| | **MorphGrower** | $\mathbf{41.50 \pm 1.02}$ | $\mathbf{389.06 \pm 13.54}$ | $\mathbf{659.38 \pm 30.05}$ | $\mathbf{0.898 \pm 0.002}$ | $82.43 \pm 1.41$ | $61.44 \pm 4.23$ |
| | LSTM → Transformers | $40.55 \pm 0.82$ | $378.98 \pm 12.21$ | $645.68 \pm 28.83$ | $0.903 \pm 0.003$ | $\mathbf{84.32 \pm 0.85}$ | $59.89 \pm 0.91$ |
| | - Local Condition | $39.33 \pm 1.02$ | $383.21 \pm 10.06$ | $641.00 \pm 23.99$ | $0.918 \pm 0.003$ | $77.30 \pm 10.86$ | $60.53 \pm 0.75$ |
| | - Global Condition | $38.12 \pm 0.26$ | $372.66 \pm 10.30$ | $613.76 \pm 33.09$ | $0.929 \pm 0.003$ | $78.29 \pm 3.19$ | $\mathbf{57.21 \pm 0.48}$ |
| | - EMA | $38.97 \pm 0.82$ | $\underline{383.77 \pm 14.04}$ | $636.61 \pm 40.45$ | $0.921 \pm 0.004$ | $80.09 \pm 3.24$ | $\underline{58.77 \pm 0.91}$ |

Therefore, simultaneous modeling of the axon and dendrite of the VPM dataset is not appropriate, and we focused solely on generating the dendrite part.

## J   MORE EXPERIMENT RESULTS

Due to the limited pages for main paper, we present more experimental results in this section. Recall that we use 'MorphGrower[†]' to represent the version that does not require the provision of soma branches, while '**MorphGrower**' represents the version where we directly provide soma branches.

### J.1   ABLATION STUDY

We conduct additional experiments to analyze the contributions of various model components to the overall performance. This included evaluating the effects of removing the local condition (Eq. 6) or the global condition (Eq. 4), as well as replacing the Exponential Moving Average (EMA) (Eq. 5) with a simpler Mean Pooling of ancestor branch representations for our local condition.

Table 9 empirically indicates that each proposed technique has effectively enhanced the performance of our model. Additionally, we experimented with replacing LSTM with Transformer and found that performance improvement was not significant, suggesting that LSTM is already sufficiently powerful for this task.

## J.2 Quantitative Results on Morphological Statistics Excluding Soma Branches

Recall that **MorphGrower** represents the version where we choose to directly give the soma branches instead of also generating them. Section 4.2 evaluates methods in terms of the overall neuronal morphologies, thus including the given reference soma branches when evaluating the performance of our method. Here, we report the quantitative results the with soma branches excluded.

**Remark.** Here, given a neuronal morphology $T$, we delete all edges on soma branches. Then the morphology $T$ is decomposed into several isolated nodes and several subtrees. Each root of the subtrees is the ending point of the original soma branches in $T$. Now, we regard each subtree as a new single neuronal morphology. Notice that the branches in the 1-th layer of the original $T$ now turn to new soma branches in those substrees.

Table 10 summarizes the quantitative results excluding the soma branches. We see that the morphologies generated by MorphGrower also well fits all six selected quantitative characterizations of given reference data.

Table 10: Quantitative Results of our MorphGrower on the VPM, RGC, M1-EXC and M1-INH datasets in terms of six quantitative metrics excluding soma branches.

| Dataset | Method | MBPL / μm | MMED / μm | MMPD / μm | MCTT / % | MASB / ° | MAPS / ° |
|---|---|---|---|---|---|---|---|
| VPM | *Reference* | $59.66 \pm 0.72$ | $117.74 \pm 1.19$ | $133.71 \pm 0.71$ | $0.928 \pm 0.001$ | $74.93 \pm 1.15$ | $29.49 \pm 0.58$ |
| | **MorphGrower** | $55.53 \pm 0.58$ | $116.17 \pm 1.67$ | $125.72 \pm 1.78$ | $0.920 \pm 0.005$ | $82.96 \pm 2.01$ | $36.00 \pm 1.13$ |
| RGC | *Reference* | $27.03 \pm 0.81$ | $212.73 \pm 5.61$ | $276.53 \pm 7.72$ | $0.936 \pm 0.001$ | $141.09 \pm 1.47$ | $49.36 \pm 0.67$ |
| | **MorphGrower** | $25.56 \pm 0.80$ | $210.87 \pm 5.34$ | $261.60 \pm 7.30$ | $0.945 \pm 0.001$ | $138.23 \pm 2.30$ | $50.08 \pm 0.90$ |
| M1-EXC | *Reference* | $67.43 \pm 1.81$ | $168.29 \pm 3.07$ | $198.03 \pm 3.03$ | $0.887 \pm 0.004$ | $68.39 \pm 2.64$ | $30.56 \pm 1.25$ |
| | **MorphGrower** | $61.79 \pm 1.23$ | $166.68 \pm 3.93$ | $184.85 \pm 4.12$ | $0.930 \pm 0.003$ | $68.83 \pm 1.91$ | $31.84 \pm 1.42$ |
| M1-INH | *Reference* | $58.54 \pm 1.88$ | $181.32 \pm 7.84$ | $254.79 \pm 11.23$ | $0.888 \pm 0.002$ | $89.06 \pm 4.16$ | $36.39 \pm 2.01$ |
| | **MorphGrower** | $54.28 \pm 1.62$ | $177.74 \pm 7.33$ | $237.60 \pm 9.99$ | $0.916 \pm 0.004$ | $88.37 \pm 3.59$ | $41.78 \pm 4.92$ |

Table 11: Classification accuracy (%). The closer accuracy to 50% indicates higher generation plausibility. The best and the runner-up plausibile in each columns are highlighted in **bold** and underline respectively.

| Method \ Dataset | **VPM** | **RGC** | **M1-EXC** | **M1-INH** |
|---|---|---|---|---|
| **MorphVAE** | $86.75 \pm 06.87$ | $94.60 \pm 01.02$ | $80.72 \pm 10.58$ | $91.76 \pm 12.14$ |
| **MorphGrower** | $\mathbf{54.73 \pm 03.36}$ | $\underline{62.70 \pm 05.24}$ | $\mathbf{55.00 \pm 02.54}$ | $\mathbf{54.74 \pm 01.63}$ |
| **MorphGrower**[†] | $\underline{59.47 \pm 04.53}$ | $\mathbf{54.99 \pm 03.08}$ | $\underline{55.26 \pm 02.42}$ | $\underline{55.47 \pm 03.43}$ |

Table 12: Results of the average BlastNeuron Distance on RGC.

| Method \ Reference | $T_1$ | $T_2$ | $T_3$ | $T_4$ | $T_5$ |
|---|---|---|---|---|---|
| **MorphVAE** | 15109.85 | 15358.16 | 12591.39 | 11898.83 | 40403.69 |
| **MorphGrower** | 384673.12 | 55149.68 | 134330.53 | 65100.22 | 938817.33 |

Table 13: Results of the average BlastNeuron Distance on M1-EXC.

| Method / Reference | $T_1$ | $T_2$ | $T_3$ | $T_4$ | $T_5$ |
|---|---|---|---|---|---|
| **MorphVAE** | 1392.08 | 5071.27 | 2177.73 | 12477.61 | 2011.50 |
| **MorphGrower** | 97910.19 | 48697.82 | 19896.38 | 34297.98 | 45347.52 |

Table 14: Results of the average BlastNeuron Distance on M1-INH.

| Method / Reference | $T_1$ | $T_2$ | $T_3$ | $T_4$ | $T_5$ |
|---|---|---|---|---|---|
| **MorphVAE** | 1174.03 | 4086.11 | 16625.34 | 35649.40 | 4671.48 |
| **MorphGrower** | 29634.89 | 43372.90 | 119178.31 | 207115.29 | 69570.35 |

## J.3 PLAUSIBILITY EVALUATION FOR THE SAMPLES GENERATED WITHOUT SOMA BRANCHES GIVEN

We conducted an additional evaluation experiment regarding the "plausibility" of generated morphology samples without soma branches directly given. The results are presented in Table 11. Based on the table, it is evident that MorphGrower and MorphGrower$^\dagger$ perform similarly in this evaluation, making it challenging to decisively determine which one exhibits superior performance. However, both MorphGrower and MorphGrower$^\dagger$ consistently generate samples with higher plausibility compared to the baseline MorphVAE.

## J.4 MORE GENERATION DIVERSITY EVALUATION

In the main paper, we only evaluate the generation diversity on VPM dataset. Here, we present evaluation on the other three datasets: RGC, M1-EXC and M1-INH.

We also randomly pick five neuronal morphologies from the test set of RGC, M1-EXC and M1-INH datasets as references, respectively. For each reference, we generate 50 corresponding samples using both MorphVAE and MorphGrower. Then, we also report the average BND over five selected morphologies on each dataset. Results are demonstrated in Table 12, Table 13 and Table 14. Results show that the average BNDs of morphologies generated by MorphGrower are consistently higher on the other three datasets, indicating that the diversity of samples generated by MorphGrower is greater than the MorphVAE.

## J.5 SENSITIVITY TO THE EMBEDDING SIZE

In this section, we investigate the sensitivity of our method and the baseline MorphVAE to the embedding size.

Figure 12 illustrates the performance of baseline MorphVAE and our MorphVAE on six metrics used in Sec. 4.2 as the embedding size varies. We selected four embedding sizes: $\{16, 32, 64, 128\}$. It can be observed that MorphVAE's performance on these four embedding size choices is inferior to MorphGrower with the same embedding size.

MorphGrower shows an improvement in performance in metrics such as MBPL, MMED, MMPD, and MCTT as the embedding size increases, but then it decreases as the embedding size further increases. This might be due to the limited amount of data, making it challenging to fit a larger model. On the other hand, in metrics like MASB and MAPS, the model's performance improves as the embedding size increases for these four different choices. This is likely because larger models help capture complex patterns and relationships related to angle correlations in the data.

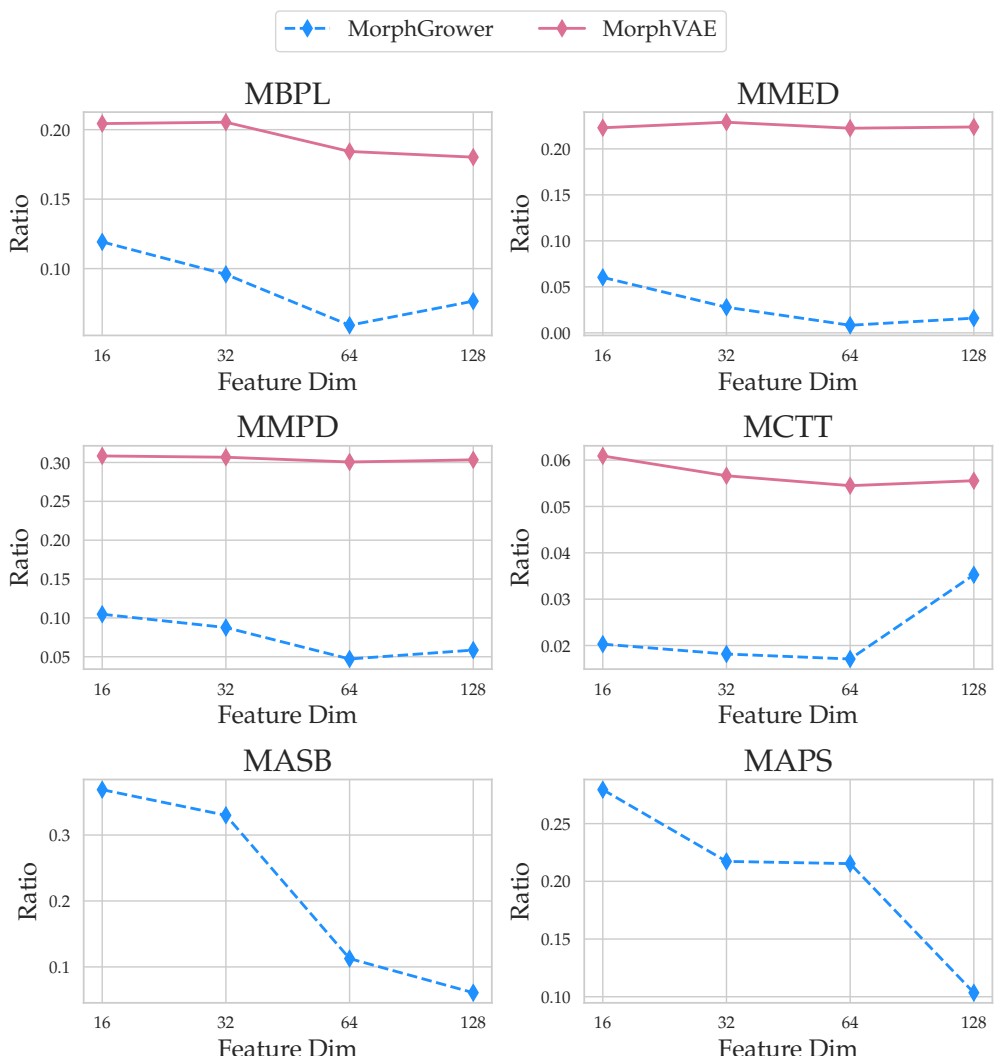

Figure 12: Performance curves of MorphVAE and MorphGrower as the embedding size varies, in terms of MBPL, MMED, MMPD, MCTT, MASB, and MAPS. The horizontal axis represents the embedding size, while the vertical axis indicates the ratio of the deviation from real data metrics to the real data metrics themselves. A lower ratio indicates better performance. We leave MorphVAE's curves on MBPL and MAPS blank because it may generate nodes with more than two subsequent branches that conflict with the definition of MBPL and MAPS for bifurcations.

## J.6 EVALUATION IN TERMS OF FID AND KID

FID (Fréchet Inception Distance) and KID (Kernel Inception Distance) are commonly used as evaluation metrics for image generation methods. However, it's important to note that they typically require feature extraction from samples before computation, which is often done using pre-trained networks like ResNet (He et al., 2016) in the realm of computer vision. Yet, neuronal morphology generation isn't an image generation task, and there aren't any pre-trained extractors available for neuron morphology data on extensive datasets. Therefore, strictly speaking, we cannot provide convincing results based on FID and KID.

Nevertheless, we still try our best to present some results from the perspectives of FID and KID. Recalling our main content in Sec. 4.3, we utilize the network architecture displayed in Fig. 11 to train a real *vs.* fake classifier to evaluate the plausibility of generated samples. Here, we continue to consider leveraging computer vision techniques to extract features for morphologies. Please refer to Appendix H.3 for our rationale behind this approach. We adopt the network architecture from Fig. 11 to train a feature extractor. Initially, we set up an experiment to further verify if such an architecture indeed can learn useful representations: As neurons from different categories (i.e., from various datasets) have different morphological features, we combine real samples from the four datasets. Using the dataset as a label, our goal is to train a ResNet18-based model for classification. We conduct this experiment with diverse data splits five times at random, and the mean and variance of the classification accuracy across these results are 0.9188 and 0.0220, respectively. Clearly, this model can acquire useful representations that generalize to unseen data.

Subsequently, we employ the newly trained ResNet18-based model as a feature extractor to compute results on both KID and FID metrics. Table 15 showcases the results in terms of KID and FID.

All results were averaged from five experiments. From the results, we can observe that whether it's the KID or FID metric, our method significantly outperforms the baseline MorphVAE across all datasets. Because we adopted the evaluation method of image generation here, it further confirms that the samples generated by our method are visually closer to real samples compared to those produced by the baseline MorphVAE.

Table 15: Evaluation of generation performance on the VPM, RGC, M1-EXC and M1-INH datasets in terms of FID and KID.

| Metric | Method | VPM | RGC | M1-EXC | M1-INH |
|--------|--------|-----|-----|--------|--------|
| FID | MorphVAE | $85.95 \pm 5.13$ | $208.42 \pm 4.39$ | $168.63 \pm 9.82$ | $1651.17 \pm 91.51$ |
| | MorphGrower | $\mathbf{23.71 \pm 5.06}$ | $\mathbf{106.16 \pm 8.33}$ | $\mathbf{17.07 \pm 5.73}$ | $\mathbf{62.37 \pm 14.38}$ |
| KID | MorphVAE | $5.83 \pm 0.76$ | $22.65 \pm 1.84$ | $7.86 \pm 0.73$ | $356.67 \pm 22.07$ |
| | MorphGrower | $\mathbf{1.48 \pm 0.34}$ | $\mathbf{9.42 \pm 0.85}$ | $\mathbf{0.10 \pm 0.12}$ | $\mathbf{19.87 \pm 7.17}$ |

## K NOTATIONS

We summarize the notations used in this paper in Table 16 to facilitate reading.

## L THE DISTINCT SCOPE OF MORPHGROWER *vs.* MORPHVAE

**MorphVAE** primarily addresses two main issues:

- It introduces method for augmenting neuronal morphology data using 3D walks as the fundamental generation unit.
- It proposes a method for extracting an overall representation of neuron morphology data: a pooling operation is applied to the latent embeddings corresponding to all walks. This overall representation can be applied to downstream tasks such as neuron classification.

In MorphVAE, the generation module and the downstream classifier are trained simultaneously. However, it's worth noting that MorphVAE's primary focus and contribution lie in augmenting

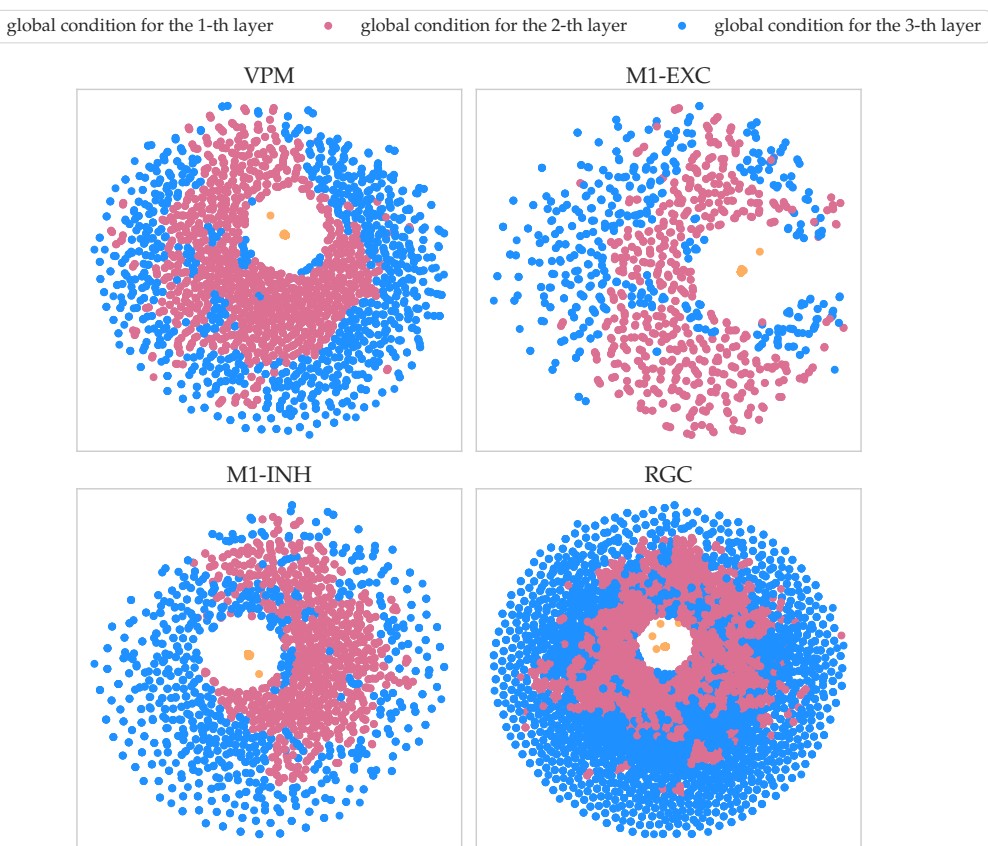

Figure 13: We use t-SNE to visualize the global conditions learned by MorphGrower on four datasets.

neuron morphology data. The extraction of overall features of neuron morphology, as described in the second point, can be considered an additional byproduct. Additionally, MorphVAE uses a relatively large step size for resampling neuron data, resulting in the loss of a substantial amount of fine-grained information and the possibility of introducing multiple branching points in the augmented data. This does not align with the reasonable requirements of neuronal morphology topology, further indicating that MorphVAE may not be a particularly suitable model for augmenting neuronal morphology data.

**MorphGrower** focuses on augmenting neuron morphology data at a finer granularity: We employ a layer-by-layer generation strategy and generate branches in pairs within each layer. We do not aim to learn an overall representation of neuronal morphology for downstream tasks. However, it is worth noting that we have found that the global condition module proposed by MorphGrower also has the ability to capture complex patterns in the data, as detailed in Appendix N.1. Additionally, we use a more reasonable data preprocessing approach by using a smaller step size for normalizing neuron samples. These measures ensure that our augmentation pipeline results in neuron data with more detail and guarantees the topological validity of generated samples.

## M POTENTIAL NEGATIVE IMPACTS

As far as we are concerned, we have not identified any negative social impact of this work.

## N VISUALIZATION

In this section, we present more visualization results.

### N.1 Visualization of the Learned Global Condition Using t-SNE

Figure 13 visualizes the global conditions learned by MorphGrower using t-SNE (Van der Maaten & Hinton, 2008). Since the morphologies in all four datasets typically include at least four layers of tree structure (requiring the extraction of global conditions three times for generating four layers), in Fig. 13, we chose to depict the global conditions extracted when generating the 1-th, 2-th, and 3-th layers (in our paper, layer indexing starts from 0). We can see that the global conditions for different layers can be well classified, indicating that MorphGrower can indeed capture certain patterns in the data. Therefore, the representations of the extracted global conditions are meaningful, which further confirms that MorphGrower can effectively model the influence of predecessor branches on subsequent branches.

### N.2 Visualization of the Distribution of Four Metrics

We only plot the distributions of MPD, BPL, MED and CTT over the reference samples and generated morphologies on the VPM dataset in the main paper due to the limited space. Here, we plot the their distributions over the other three datasets in Fig. 14, Fig. 15 and Fig. 16.

### N.3 Illustration of the Layer-by-layer Generation

We demonstrate the complete sequence of the intermediate generated morphologies and the final generated morphology for the selected example from the RGC dataset in Sec. 4.5 in Fig. 18.

Besides, we also randomly pick a generated morphology from each of the other three datasets and demonstrate the sequence of the intermediate generated morphologies and the final generated morphology for each corresponding example in Fig. 17, Fig. 19 and Fig. 20.

We see that MorphGrower can generate snapshots of morphologies in different growing stages. Such a computational recovery of neuronal dynamic growth procedure may be an interesting future research direction and help in further unravelling the neuronal growth mechanism.

### N.4 Sholl Analysis

In this study, we utilize the Sholl analysis method (Sholl, 1953) to quantitatively evaluate the complexity of morphological differences between reference and generated neurons. The Sholl intersection profile of a specific neuron is represented as a one-dimensional distribution, which is derived by enumerating the number of branch intersections at varying distances from the soma. This analytical approach enables the quantitative comparison of morphologically distinct neurons and facilitates the mapping of synaptic contact or mitochondrial distribution within dendritic arbors (Lim et al., 2015). Moreover, the Sholl analysis has proven instrumental in illustrating alterations in dendritic morphology resulting from neuronal diseases (Beauquis et al., 2013), degeneration (Williams et al., 2013), or therapeutic interventions (Packer et al., 2013).

We conduct Sholl analysis on each neural morphology data in the testing set of each dataset to investigate dendritic arborization. Prior to analysis, we standardized each morphology data by translating the soma (root node) to the origin $(0, 0, 0)$. Thereafter, we generated 20 concentric spheres with the soma as the center such that they were equidistant from one another and spanned the maximal coordinate range. We assess the number of dendritic intersections between each concentric sphere and different neuron samples and depicted the results as a plot such that the average number of intersections was depicted on the $y$-axis and the radius of the concentric sphere on the $x$-axis. The findings are represented in Fig. 21, Fig. 22, Fig. 23 and Fig. 24. The visualized distribution indicates that our method closely approximates the real samples in the testing set, whereas the performance of MorphVAE significantly deviates from them.

Additionally, we randomly select a morphology sample from each dataset and perform a Sholl analysis on samples generated by various methods, utilizing the chosen sample as a reference. The visualized outcomes are illustrated in Fig. 25, Fig. 26, Fig. 27 and Fig. 28. Notably, the samples generated by MorphVAE display a considerable divergence from the reference realistic sample, while the samples generated by both MorphGrower and MorphGrower[†] closely resemble the reference.

### N.5 MORE VISUALIZATION RESULTS OF GENERATED MORPHOLOGIES

We present more visualizations of some neuronal morphologies generated by our proposed MorphGrower in Fig. 29, Fig. 30, Fig. 31 and Fig. 32. Owing to image size constraints, we were unable to render the 3D image with uniform scaling across all three axes. In reality, the unit scales for the $x$, $y$, and $z$ axes in the 3D image differ, leading to some distortion and making the elongation in specific directions less noticeable. The 2D projection images feature consistent unit scaling for each axis, which explains why the observed neuronal morphology appears distinct from the 3D image.

Table 16: Notations.

| Notation | Description |
|---|---|
| $T$ | an neuronal morphology instance |
| $\hat{T}$ | the generated morphology |
| $\hat{T}^{(i)}$ | the top $i$ layer subtree of the final generated morphology |
| $V$ | the node set of a given neuronal morphology |
| $E$ | the edge set of a given neuronal morphology |
| $b_i$ | a branch instance |
| $\hat{b}_i$ | a generated branch using $b_i$ as reference |
| $\hat{v}_i$ | a node instance on the generated morphology |
| $b_{is}$ | the branch which starts from $v_i$ and ends at $v_s$ |
| $v_i$ | the 3D coordinates of a node instance |
| $N_v$ | the node number of a given neuronal morphology |
| $e_{i,j}$ | the directed edge beginning from $v_i$ and ending with $v_j$ |
| $\mathcal{N}^+(b_i)$ | two subsequent branches of $b_i$ |
| $\mathcal{F}$ | the forest composed of branch trees, serves as the global condition |
| $L(i)$ | the number of nodes included in $b_i$ |
| $L_i$ | the branch collection of the $i$-th layer |
| $\hat{L}_i$ | the branch collection of the generated morphology at the $i$-th layer |
| $N^k$ | the number of branches at the $k$-th layer |
| $m$ | the iteration number in Eq. 4 |
| $\mu$ | the mean of the vMF distribution |
| $\kappa$ | the variance of the vMF distribution |
| $B$ | the branch pair input |
| $C$ | the input for encoding the global condition and local contidion |
| $\mathcal{A}$ | the sequence of ancestor branches |
| $\mathbf{D}_k^{\mathcal{A}}$ | the feature extracted from the first $k$ element of $\mathcal{A}$ |
| $\mathbf{h}_i$ | the internal hidden state of the LSTM network in encoder |
| $\mathbf{c}_i$ | the internal cell state of the LSTM network in decoder |
| $\mathbf{h}_i'^{(t)}$ | the internal hidden state of LSTM network in decoder |
| $\mathbf{c}_i'^{(t)}$ | the internal hidden state of LSTM network in decoder |
| $\mathbf{x}_i$ | the high dimension embedding projected by $\mathbf{W}_{in}$ in encoder |
| $\mathbf{y}_i$ | the output of LSTM in decoder |
| $\mathbf{W}_{in}, \mathbf{W}_{in}'$ | the linear transform that projects 3D coordinate into high dimension space in encoder, decoder |
| $\mathbf{W}_{sta_1}, \mathbf{W}_{sta_2}$ | the linear transform to obtain the initial internal state for decoding the branch pair |
| $\mathbf{W}_{out}$ | the linear transform that transforms $y_i$ into 3D coordinate |
| $\mathbf{W}^{(k)}$ | the linear transform in the $k$-th iteration of GNN while encoding global condition |
| $\mathbf{r}_b$ | the encoded representation of branch $b$ |
| $\mathbf{r}_{b_i}$ | the encoded representation of a specific branch $b_i$ |
| $\mathbf{h}_{b_i}^{(k)}$ | the node feature of branch $b_i$ which is located at $k$-th layer |
| $k$ in Eq. 3 | depth in the tree (starting from 0) |
| $\mathbf{h}_{local}$ | the encoded representation of local condition |
| $\mathbf{h}_{global}$ | the encoded representation of global condition |
| $\theta$ | the learnable parameters of the encoder |
| $\phi$ | the learnable parameters of the decoder |
| $f_\theta(\cdot)$ | the encoder |
| $g_\phi(\cdot)$ | the decoder |
| $\mathbf{z}$ | the latent variable |

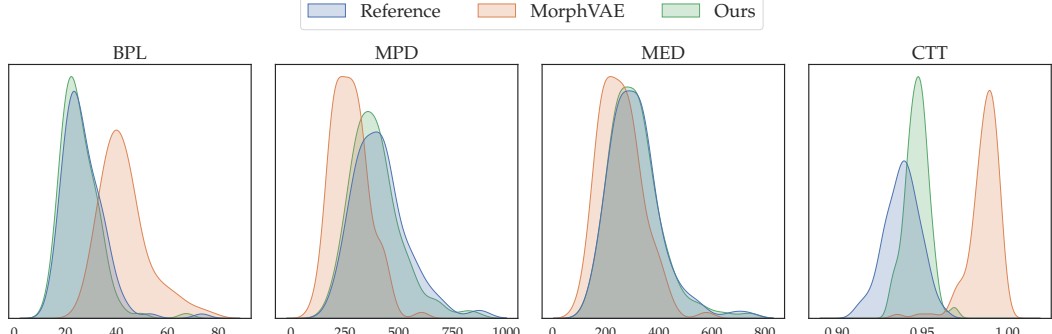

Figure 14: Distributions of the four morphological metrics: MPD, BPL, MED and CTT on the RGC dataset.

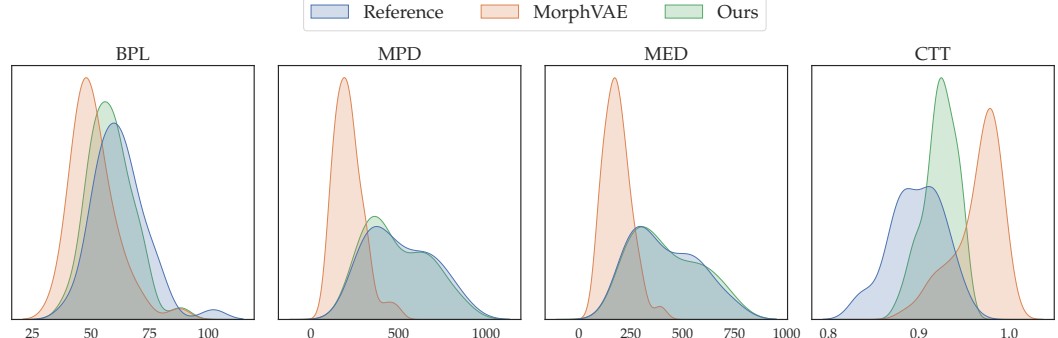

Figure 15: Distributions of the four morphological metrics: MPD, BPL, MED and CTT on the M1-EXC dataset.

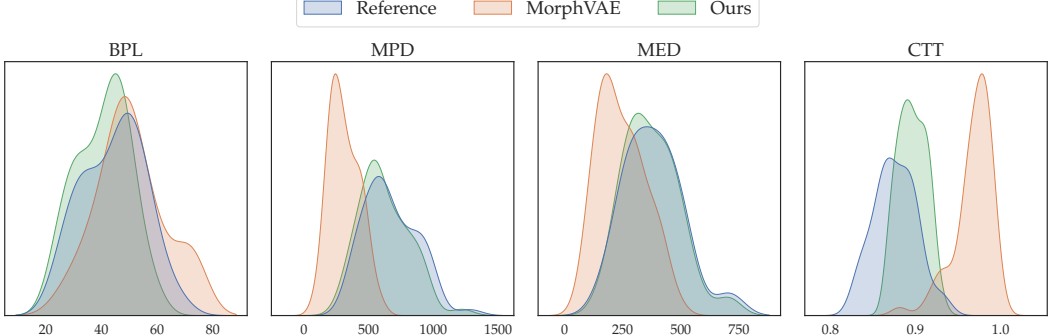

Figure 16: Distributions of the four morphological metrics: MPD, BPL, MED and CTT on the M1-INH dataset.

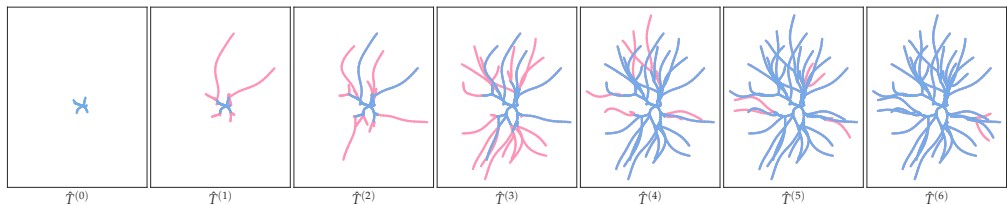

Figure 17: An example from the VPM dataset. We demonstrate the projections onto the $xy$ plane of the sequence of the intermediate morphologies $\{\hat{T}^{(i)}\}_{i=0}^5$ and the final generated morphology $\hat{T}^{(6)}$.

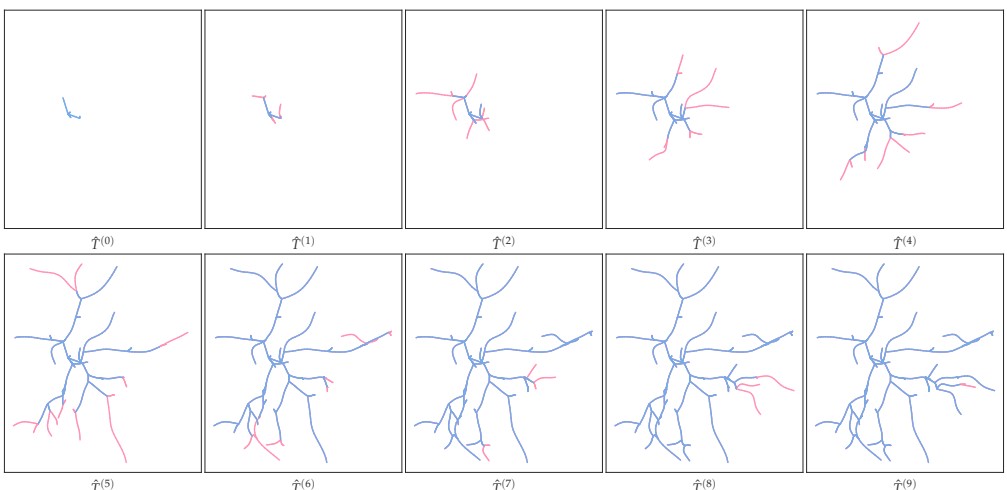

Figure 18: An example from the RGC dataset. We demonstrate the projections onto the $xy$ plane of the sequence of the intermediate morphologies $\{\hat{T}^{(i)}\}_{i=0}^8$ and the final generated morphology $\hat{T}^{(9)}$.

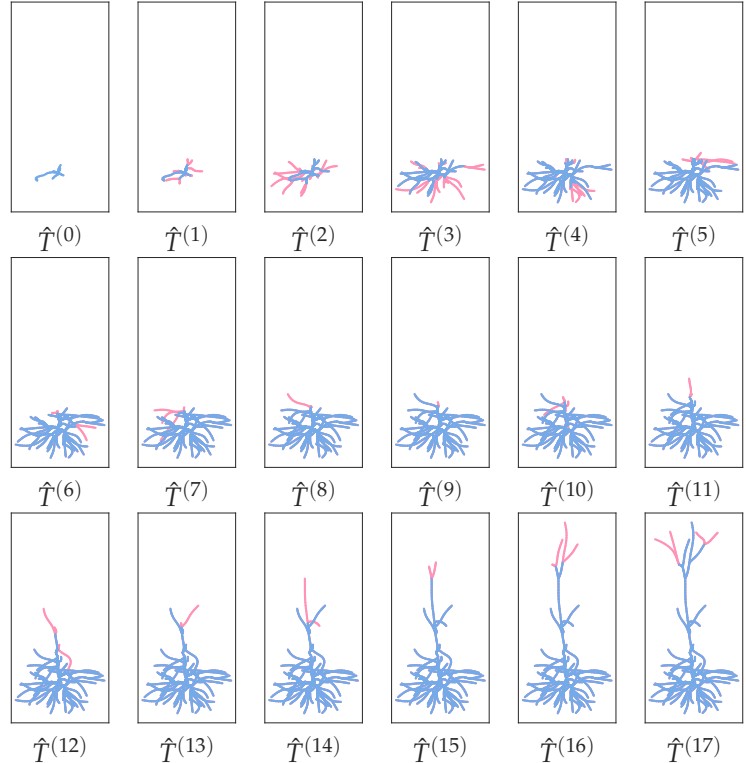

Figure 19: An example from the M1-EXC dataset. We demonstrate the projections onto the $xy$ plane of the sequence of the intermediate morphologies $\{\hat{T}^{(i)}\}_{i=0}^{16}$ and the final generated morphology $\hat{T}^{(17)}$.

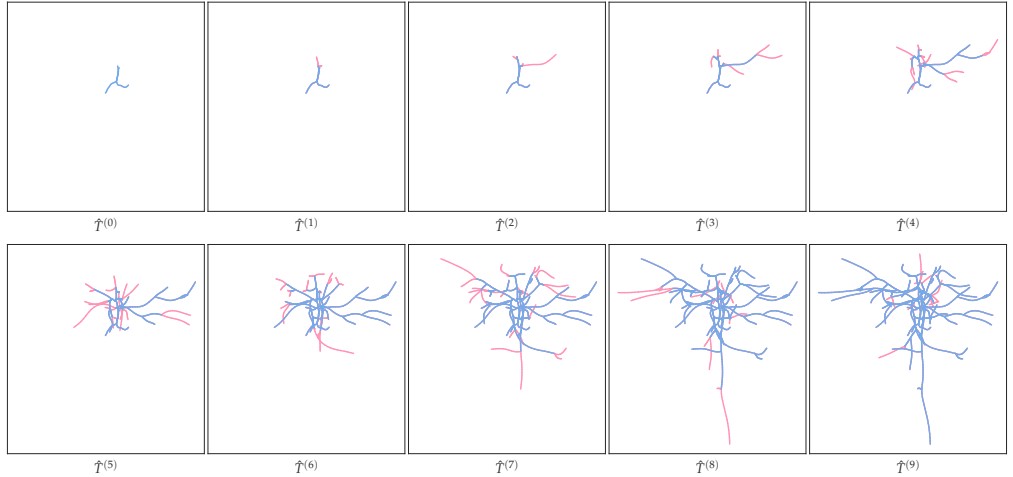

Figure 20: An example from the M1-INH dataset. We demonstrate the projections onto the $xy$ plane of the sequence of the intermediate morphologies $\{\hat{T}^{(i)}\}_{i=0}^{8}$ and the final generated morphology $\hat{T}^{(9)}$.

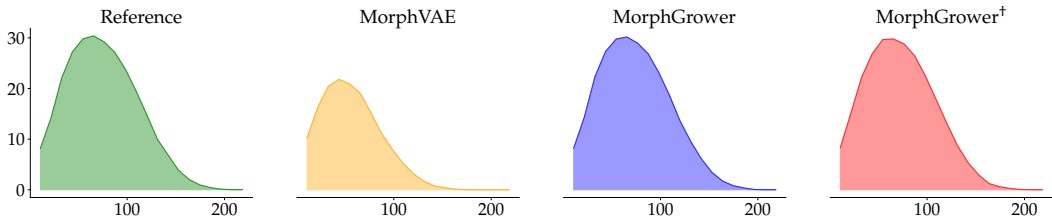

Figure 21: Visualized distribution on the VPM dataset. The horizontal axis represents the radius of the concentric spheres, and the vertical axis represents the average number of intersections between different neuron samples and the concentric spheres.

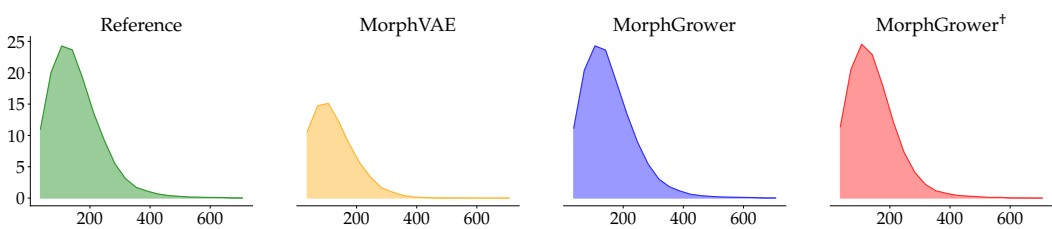

Figure 22: Visualized distribution on the RGC dataset. The horizontal axis represents the radius of the concentric spheres, and the vertical axis represents the average number of intersections between different neuron samples and the concentric spheres.

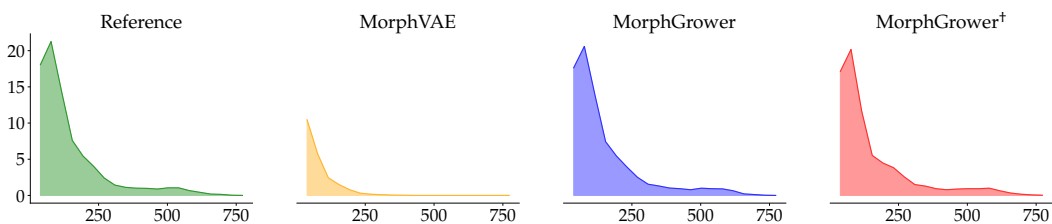

Figure 23: Visualized distribution on the M1-EXC dataset. The horizontal axis represents the radius of the concentric spheres, and the vertical axis represents the average number of intersections between different neuron samples and the concentric spheres.

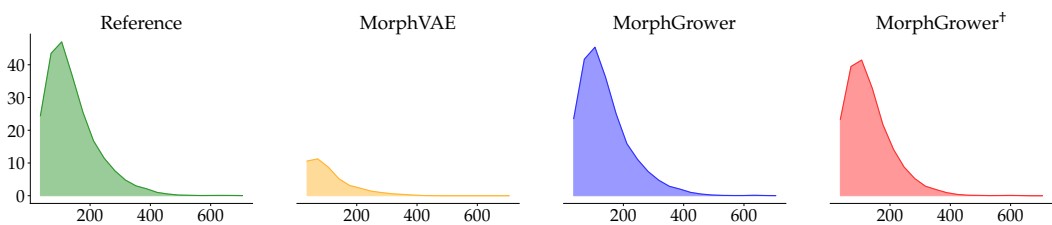

Figure 24: Visualized distribution on the M1-INH dataset. The horizontal axis represents the radius of the concentric spheres, and the vertical axis represents the average number of intersections between different neuron samples and the concentric spheres.

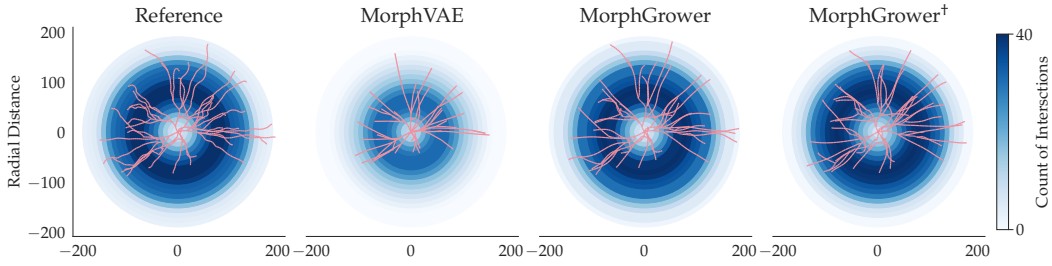

Figure 25: Visualization of Sholl analysis for a real sample from the VPM dataset and for neuronal morphologies generated by different methods.

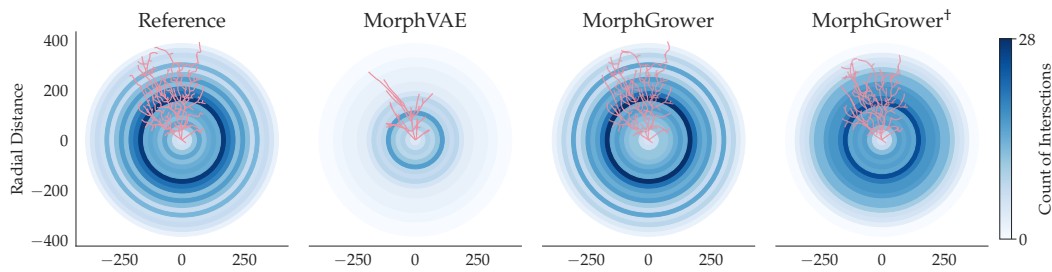

Figure 26: Visualization of Sholl analysis for a real sample from the RGC dataset and for neuronal morphologies generated by different methods

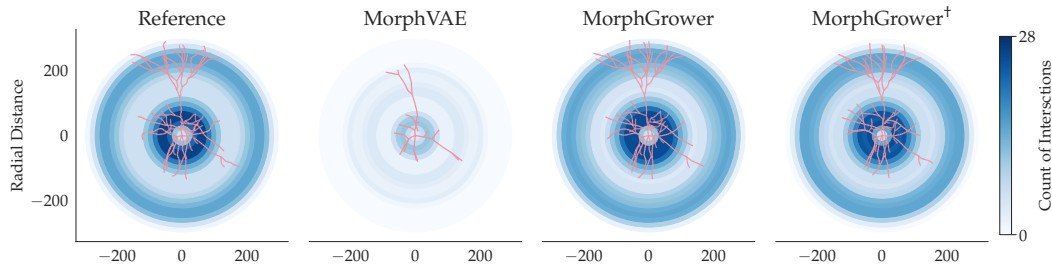

Figure 27: Visualization of Sholl analysis for a real sample from the M1-EXC dataset and for neuronal morphologies generated by different methods

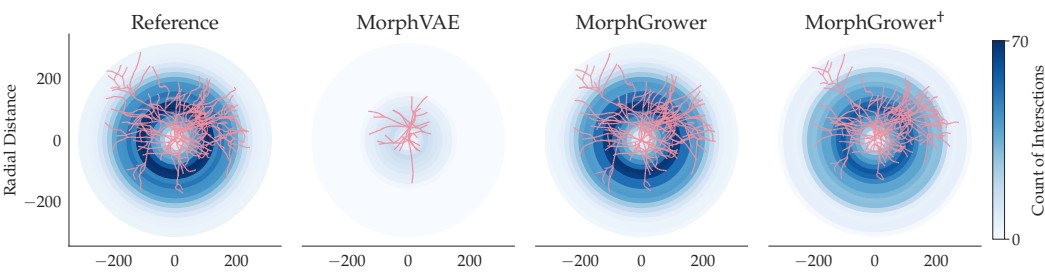

Figure 28: Visualization of Sholl analysis for a real sample from the M1-INH dataset and for neuronal morphologies generated by different methods

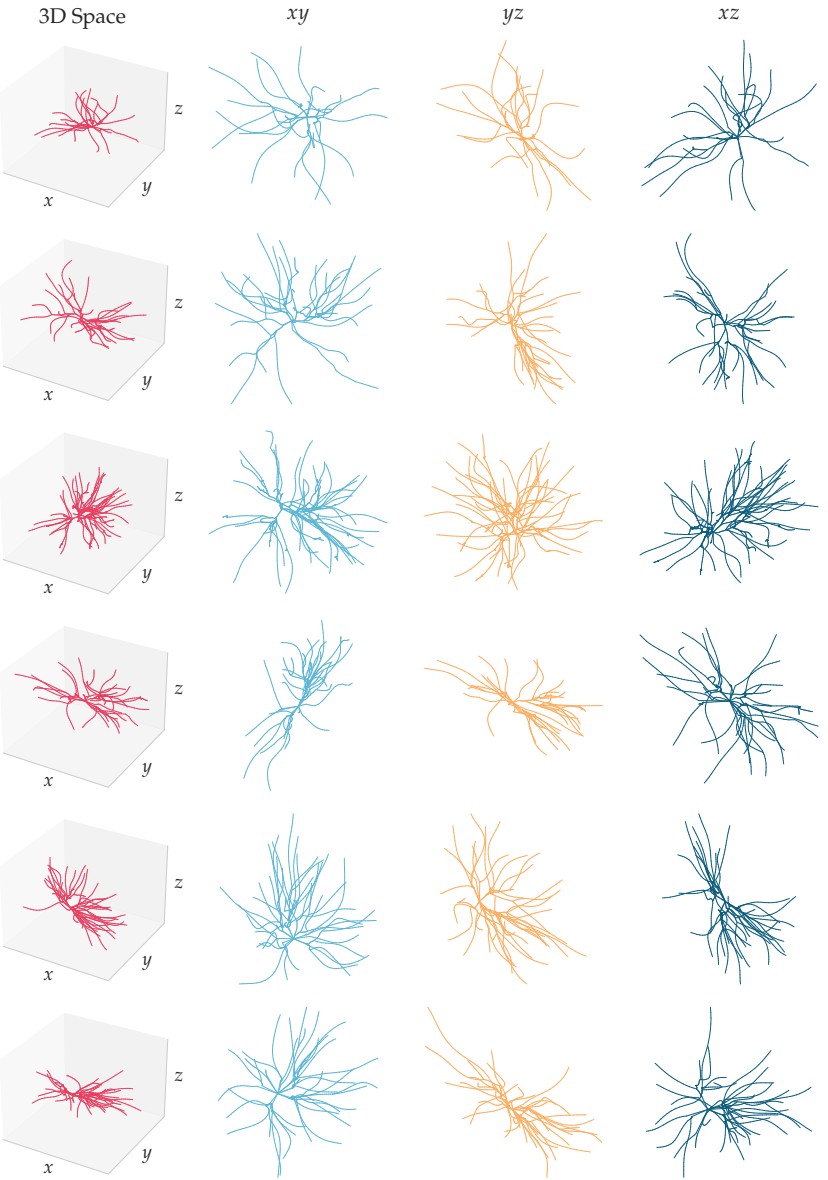

Figure 29: Visualization of some neuronal morphologies generated by **MorphGrower** over the **VPM** dataset. The first column demonstrates the morphologies in 3D space. The three columns (from left to right) on the right are the projections of morphology samples onto $xy$, $yz$ and $xz$ planes, respectively. A group of pictures on each row corresponds to a generated neuronal morphology.

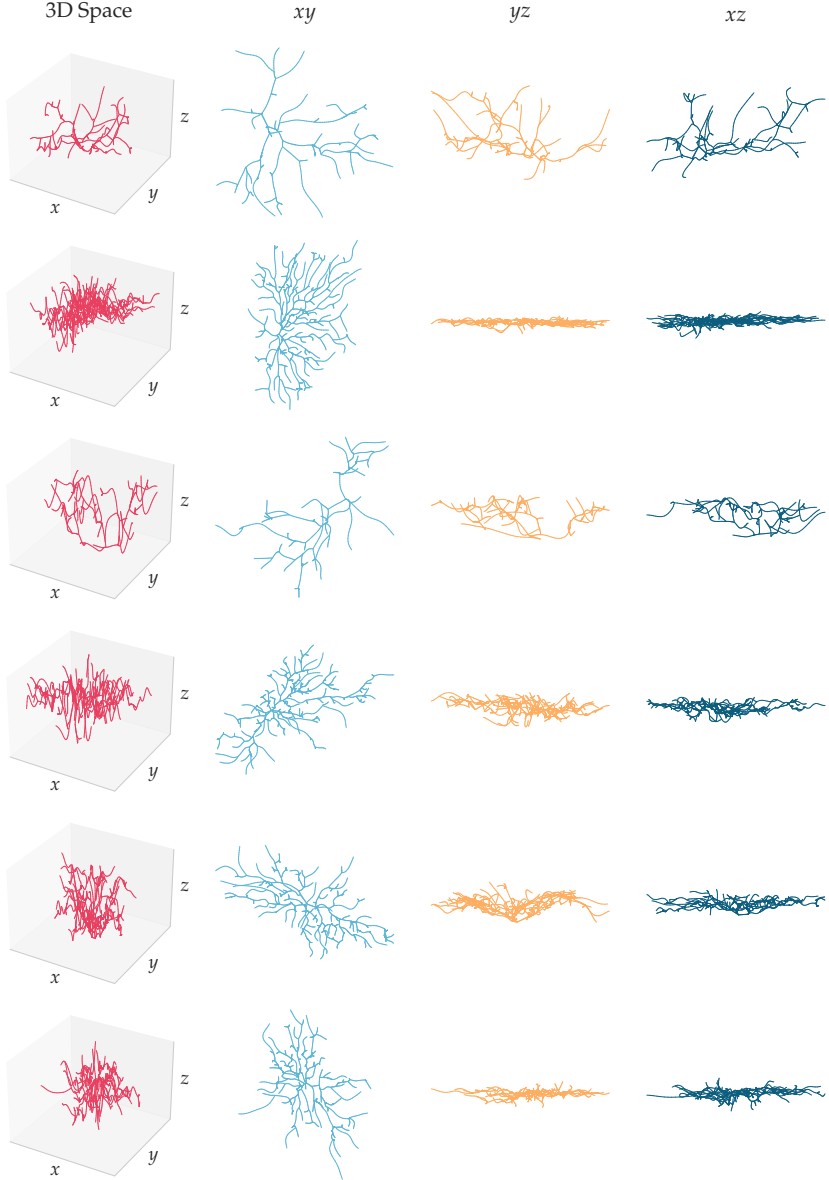

Figure 30: Visualization of some neuronal morphologies generated by **MorphGrower** over the **RGC** dataset. The first column demonstrates the morphologies in 3D space. The three columns (from left to right) on the right are the projections of morphology samples onto $xy$, $yz$ and $xz$ planes, respectively. A group of pictures on each row corresponds to a generated neuronal morphology.

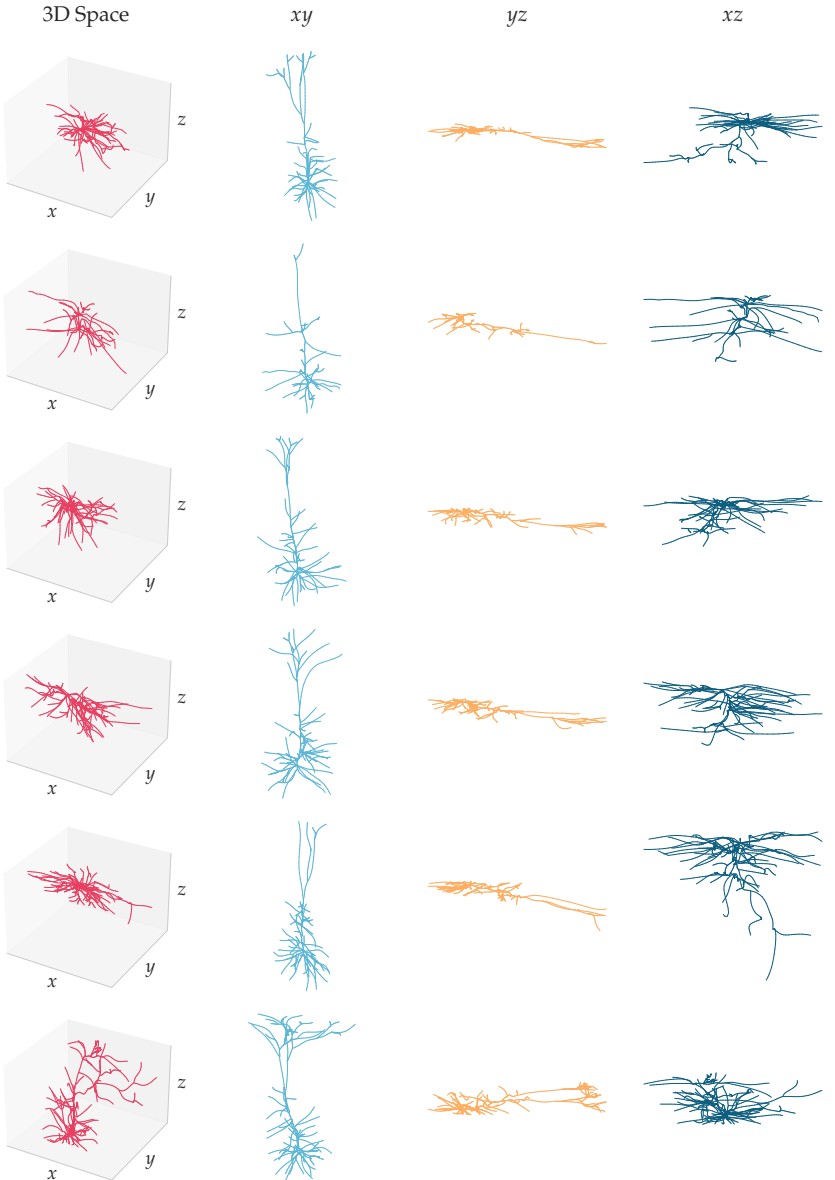

Figure 31: Visualization of some neuronal morphologies generated by **MorphGrower** over the **M1-EXC** dataset. The first column demonstrates the morphologies in 3D space. The three columns (from left to right) on the right are the projections of morphology samples onto $xy$, $yz$ and $xz$ planes, respectively. A group of pictures on each row corresponds to a generated neuronal morphology.

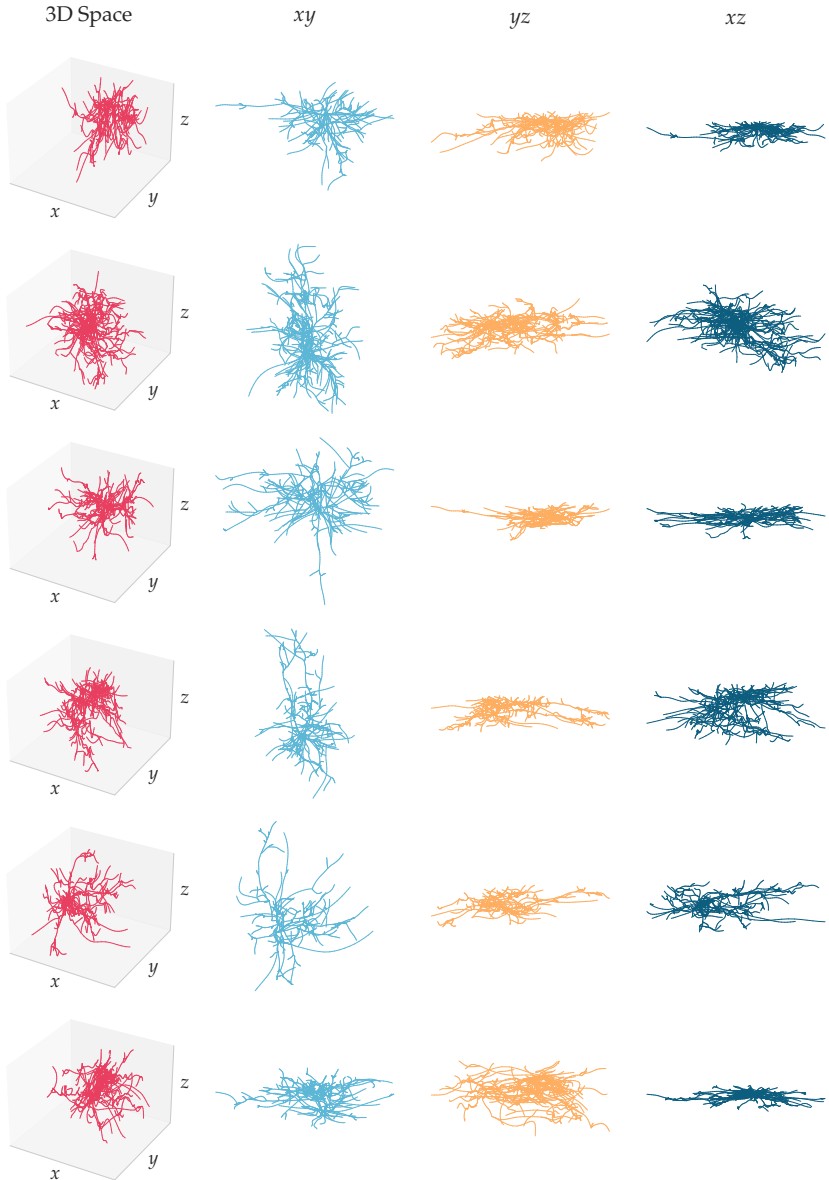

Figure 32: Visualization of some neuronal morphologies generated by **MorphGrower** over the **M1-INH** dataset. The first column demonstrates the morphologies in 3D space. The three columns (from left to right) on the right are the projections of morphology samples onto $xy$, $yz$ and $xz$ planes, respectively. A group of pictures on each row corresponds to a generated neuronal morphology.

