# OpenReview forum: "MorphGrower: A Synchronized Layer-by-layer Growing Approach for Plausible and Diverse Neuronal Morphology Generation"
_ICLR.cc/2024/Conference — Submitted to ICLR 2024_

### Official Review · Reviewer_NzbM · 2023-10-31

**Soundness:** 3 good
**Presentation:** 2 fair
**Contribution:** 2 fair
**Rating:** 5
**Confidence:** 4

**Summary:**

The paper proposes a generative model for neuronal morphologies (in skeleton form), implemented as a conditional VAE with an LSTM encoder/decoder. Neurons are grown using an autoregressive sampling procedure inspired by natural growth processes. The proposed model is shown to consistently perform better than MorphVAE (the only other deep learning-based alternative) under multiple metrics and for different tissue samples.

**Strengths:**

- The idea to generate branches progressively is interesting and makes intuitive sense.
- Terminology is clearly defined and illustrated in Sec. 2.
- The authors promise to release the source code of their method.
- Evaluations show consistently improved results in comparison to MorphVAE.
- In addition to morphological metrics, a classifier-based approach is used to verify plausibility and BND to evaluate diversity of the generated morphologies.

**Weaknesses:**

- Ablations of MAE and local/global conditioning are relegated to an appendix and restricted to a single dataset. Please consider including and discussing them in the main text. Perhaps some of the formulas for the LSTMs could be moved to the appendices to make space for this.

**Questions:**

- The importance of neuronal morphology for diseases such as Alzheimer's feels out of place in the abstract. The statement itself is true of course, but it's unclear how having a computational model of such morphologies would make studying these diseases any easier. I suggest replacing it with some other potential applications.
- In the global condition, are the non-branch coordinates completely discarded or still used somehow?
- In section 3.3 describing the sampling procedure a "reference morphology T" is described. Does that mean that the sampled neurons will always have the exact same tree structure as T?
- Have you performed any studies on the impact of the embedding size? (both for your model and MorphVAE). This hyperparameter does not appear to be explored much in the text.
- Does Fig. 4 suggest overfitting? How is it possible that the generated neurons remain so close to the original morphology?

---

> ### Author Response · Authors · 2023-11-17
> **Response to Reviewer NzbM (1/2)**
>
> Thank you for your feedback. Here are our responses to your questions.
>
> > **Response to the mentioned weaknesss.**
>
> - Firstly, we would like to humbly point out to the reviewer that we did not restrict the ablation study to a single dataset. Instead, in the appendix of the submitted manuscript, we provided ablation study results on four different datasets (Tables 11, 12, 13, and 14 in the previous version). We believe it might have been due to the formatting and separation of the four tables that led you to notice the ablation study results for only one dataset, giving the impression that we conducted experiments on only one dataset. In the latest revised version, we have consolidated the ablation study results on multiple datasets into a single table (Table 11 in the new version).
>
> - Furthermore, we appreciate your suggestion regarding the importance of ablation studies and the necessity to present them in the main text. We are aware of the significance of this aspect, which is why in the previous version submitted, in the first paragraph of Sec. 4 (the experimental section), we informed readers that we had conducted an ablation study and explicitly directed them to a specific section in the appendix for further details. However, including the results of the ablation study in the main text would require a significant amount of space, and given the limitation of a maximum of 9 pages for the main text, we opted to place the results of the ablation study in the appendix.
>
>
> > **Necessity of the importance of neuronal morphology for diseases.**
>
> - Systematic, large-scale and high-throughput acquisition of neuronal morphology can help us to understanding brain anatomy at fine scales, and has received worldwide attentions[1-3]. Such digital morphologies make a lot of downstream computational neuroscience applications possible, e.g. neuron simulation [4], classification [5], network analysis[6], studying the connection to computational models in deep learning [7], etc. Additionally, alterations in dendrite morphology influence not only dendritic function and signal transmission but also affect electrophysiological properties and neural network behavior. Abnormal dendritic morphology has been linked to brain disorders, including mental retardation [8], schizophrenia [9], autism [10], and stress-related disorders [11-13]. However, the current process of annotating morphological data requires a significant amount of manpower, which comes with high costs. This is exactly why we are developing methods for synthesizing neuronal morphologies.
>
> - Despite the aforementioned, we are very willing to accept your suggestion to omit this section of the discussion and allocate this space to other content.
>
> > **Reference:**
>
> [1] Winnubst, J. et al. Reconstruction of 1,000 Projection Neurons Reveals New Cell Types and Organization of Long-Range Connectivity in the Mouse Brain. Cell 179, 268-281.e13 (2019).
>
> [2] Peng, H. et al. Morphological diversity of single neurons in molecularly defined cell types. Nature 598, 174–181 (2021).
>
> [3] Gao, L. et al. Single-neuron projectome of mouse prefrontal cortex. Nat Neurosci 25, 515–529 (2022).
>
> [4] Markram, H. et al. Reconstruction and Simulation of Neocortical Microcircuitry. Cell 163, 456–492 (2015).
>
> [5] Gouwens, N. W. et al. Classification of electrophysiological and morphological neuron types in the mouse visual cortex. Nat Neurosci 22, 1182–1195 (2019).
>
> [6] Gao, L. et al. Single-neuron analysis of dendrites and axons reveals the network organization in mouse prefrontal cortex. Nat Neurosci 1–16 (2023).
>
> [7] Beniaguev, D., Segev, I. & London, M. Single cortical neurons as deep artificial neural networks. Neuron 109, 2727-2739.e3 (2021).
>
> [8] Kaufmann, Walter E., and Hugo W. Moser. "Dendritic anomalies in disorders associated with mental retardation." Cerebral cortex 10.10 (2000): 981-991.
>
> [9] Glausier, Jill R., and David A. Lewis. "Dendritic spine pathology in schizophrenia." Neuroscience 251 (2013): 90-107.
>
> [10] Phillips, Mary, and Lucas Pozzo-Miller. "Dendritic spine dysgenesis in autism related disorders." Neuroscience letters 601 (2015): 30-40.
>
> [11] Shansky, Rebecca M., and John H. Morrison. "Stress-induced dendritic remodeling in the medial prefrontal cortex: effects of circuit, hormones and rest." Brain research 1293 (2009): 108-113.
>
> [12] Dioli, Chrysoula, et al. "Chronic stress triggers divergent dendritic alterations in immature neurons of the adult hippocampus, depending on their ultimate terminal fields." Translational Psychiatry 9.1 (2019): 143.
>
> [13] Sandini, Corrado, et al. "Pituitary dysmaturation affects psychopathology and neurodevelopment in 22q11. 2 Deletion Syndrome." Psychoneuroendocrinology 113 (2020): 104540.

---

> ### Author Response · Authors · 2023-11-17
> **Response to Reviewer NzbM (2/2)**
>
> > **Question about the global condition.**
>
> - We extracted the global condition from all previously generated layers, as detailed in Section 3.2 of the paper. Specifically, we treated each branch generated in the earlier layers as a node. We used an LSTM to encode the branches, obtaining representations that served as the initial features for each node. The input to the LSTM consists of the coordinate information of all points constituting a branch in the spatial domain. In other words, we utilized the coordinate information of all points contained in the branches generated in the earlier layers to derive the global condition.
>
>
> > **Question about the given reference morphology.**
>
>
> - The generated neuron indeed has the exact same tree structure as the real morphology used as a reference.
>
> - This question is closely related to the next question after the following one. Here, we provide a comprehensive further explanation:
>
>    - From the original MorphVAE paper, it is evident that MorphVAE also requires a real sample as a reference when generating a new sample:
>         - The formulation of the neuronal morphology generation task in the original MorphVAE paper at Sec 2.4 is as follows: $$\hat{M}\_T=g_\phi\left(f_\theta\left(M_T\right)\right),$$
>           where $M_T$ is a set of 3D-walks that make up the morphology, which MorphVAE uses to represent a morphology. $f_\theta$ and $g_\phi$ represent the encoder and decoder, respectively. $\hat{M}_T$ represents the final generated 3D-walks.
>         - If there was no need for a reference during generation, there would not be a test dataset. However, the original MorphVAE paper clearly defines train-valid-test dataset splits.
>
>    - In our paper, we adopt the same experimental setting as MorphVAE. This setting is somewhat similar to "augmentation", and we have justified the validity of such a setting in the introduction of our paper (refer to the the fourth paragraph in Section 1 for both the old and revised versions).
>
> - Now, returning to the question itself, our method adopts a layer-by-layer generation strategy and generates branches in pairs within each layer. This approach allows us to strictly adhere to the same tree structure as the reference. In contrast, when MorphVAE clusters the generated 3D-walks, it can create forks with more than two subsequent branches (see the Appendix C.1 for both the old and revised versions of the paper). This contradicts the basic rules of neuronal growth and the fundamental requirements of neuronal morphology data, which somewhat goes against the purpose of the augmentation task. Therefore, MorphVAE lacks internal consistency.
>
>
>
> > **Sensitivity to the embedding size.**
>
> - We have provided the results of how the performance of our method, MorphGrower, and baseline MorphVAE is affected by changes in the embedding size. However, due to time constraints, we have only presented experimental results on the VPM dataset. The corresponding results and analysis have been added to Appendix J.5 in the revised manuscript.
>
>
> > **Question about over-fitting.**
>
> - Referring to our response in the question before the previous one, since we, like the baseline MorphVAE, generate new neurons in an augmentation-like manner, it is reasonable for the generated morphology to closely resemble the given reference morphology, and this does not indicate over-fitting.

---

> > ### Comment · Reviewer_NzbM · 2023-11-22
> >
> > Thank you for all the responses and clarifications -- these are very helpful!
> >
> > I think the paper would benefit a lot from being very upfront about the importance of the template morphology (i.e. what you explained in your responses above). You already mention this in Sec. 1, but it's easy to miss when one does not know what one is looking for. It also does not explicitly say what is the nature of the "augmentation" -- i.e. that the topology stays exactly the same, but the detailed positions of the nodes might change.
> >
> > While this is very similar to the MorphVAE baseline to which you compare, in my opinion this is a fairly severe limitation of the overall approach as it very strongly constrains the diversity of shapes that might be generated (which then also limits any potential downstream applications; for instance, it's still unclear to me how this could be used for studying diseases). A "growing approach for morphology generation" evokes images of a neuron extending its branches from the soma to establish its final shape. As is, I think many readers might miss the fact that in the growing process a preexisting neuron morphology is being followed closely, making the final result more of a perturbation, rather that de-novo growth.

---

> > > ### Author Response · Authors · 2023-11-22
> > > **Sincere Gratitude to the Reviewer**
> > >
> > > Thank you sincerely for your response and the valuable suggestions you have provided. We are committed to diligently refining our paper based on your feedback.

---

### Official Review · Reviewer_7LXV · 2023-11-01

**Soundness:** 2 fair
**Presentation:** 1 poor
**Contribution:** 2 fair
**Rating:** 3
**Confidence:** 3

**Summary:**

- S1. MorphVAE encoded sequences of arbitrary number of consecutive vertices along a branch with an LSTM to learn a fixed length representation. It also used an LSTM to generate a new branch conditioned on that representation.

 - S2. As I understand, the proposed neuron generation method is:

    1. initialize `active_vertices` with soma vertex
    2. at each step, generate two branches, conditioned on already generated graph
    3. generated branches are allowed to be null.
    4. replace `active_vertices` with tips of non-null generated branches
    5. repeat 2 $\rightarrow$ 4 until `active_vertices` is empty

 - S3. Conditioning on the already generated graph requires a fixed length representation of a graph. Authors propose to do this in 2 ways:

    1. _global context_ aggregates fixed length branch-level representations at different branch orders $\dagger$

    2. _local context_ uses a discount scheme to weigh contribution at different branch orders.

 - S4. MorphVAE combine "walks" (sampled branches) with a heuristic procedure to construct a neuron tree.

 - S5. This manuscript uses a similar scheme to generate branches, but proposes a recursive procedure to generate the neuron tree.

 - S6. Authors provide a comparison of trees generated by either method based on 4 datasets.

**Strengths:**

- Manuscript builds on ideas in MorphVAE and proposes a meaningful extension.
 - The auto-regressive scheme of generating morphologies is interesting, and perhaps a good direction to think about the problem of generative models for neurons.

**Weaknesses:**

- W1. The writing and notation would benefit from being more concise and self-contained. For example, various metrics are only referred to by their acronyms in the main text. Separating the biological motivation/justification in a single paragraph instead of describing it after each method step would also help towards this end. Some symbols aren't introduced at all.

 - W2. The proposed method seems to be so over-fit to training data, that the morphologies hardly differ from the given sample (e.g. examples in Fig 4a-c., and also Fig. 23-26)? Is this not a major problem?

 - W3. Building on W2., consider a _model_ that simply jitters branch points of the reference morphology by a small amount. Based on evaluations presented in the manuscript, this procedure would
    - obtain near-perfect match on the metrics in Table on p.7.
    - match distributions in Fig. 3
    - be hardest to distinguish for classifiers (Sec. 4.3)
    - have higher BlastNeuron distances with simple heuristics (e.g. deletion of small fraction of terminal branches)

    From the manuscript and the metrics chosen to justify the generated morphologies, it is unclear to me why one should prefer the proposed method over this simple procedure to generate morphologies.

 - W4. The language is often not careful; some strong biological claims are made that are not well substantiated. Examples:

    > Since the neuronal morphology is static, the generation order of the branch pairs in each layer does not matter.

    > Remark. A typical neuron is comprised of a soma, dendrites, and an axon. Dendrites perform a random walk from the soma, while the elongation and bifurcation of axons exhibit specificity.

**Questions:**

- Q1. If the intention is to also use such models to study morphology as it relates to neuronal development, the following view should probably be reconsidered?
    > Since the neuronal morphology is static, the generation order of the branch pairs in each layer does not matter.

 - Q2. It looks like a typo (should be 3 instead of 3k), but just to make sure please clarify:
    > Most nodes on the tree have no more than 3k k-hop neighbors, thereby limiting the receptive field of nodes.

 - Q3. I assume $k$ here refers to the depth of the tree?
    > For a branch $b_i$, its corresponding node feature at the k-th iteration $\hat{h}(k)$ is calculated by

 - Q4. $\textbf{r}_{b_i}$ is not defined in the section on global context.

---

> ### Author Response · Authors · 2023-11-17
> **Response to Reviewer 7LXV (1/3)**
>
> Thank the reviewer for the time and the valuable comments. We hope that the following responses will help to address your concerns.
>
>
>
> > **Response to Q1.**
> - Assuming that we have generated up to the $i$-th layer, using all previously created layers as a condition. In the $i$-th layer, each pair of branches possesses two conditions: global and local. The global condition ensures consistency within the layer, while the local condition varies among different pairs, only depending on their respective ancestor branches. This approach guarantees that the generation order of branch pairs within the same layer does not have any impact, enabling synchronous generation.
>
> - Regarding the sentence you mentioned, we intended to convey that our focus in neuronal morphology generation is on generating a static morphology, as the original data consists solely of such static morphologies, lacking data on the dynamic growth process. We drew inspiration from the influence of previous branches on subsequent branch generation during neuronal growth, rather than attempting to directly simulate the growth process itself or what you referred to as neuronal development. If the intention were to simulate the growth process, the sentence you pointed out would indeed be incorrect.
>
> - To avoid any ambiguity caused by this sentence, we have removed it and added the detailed explanation mentioned above in the revised version.
>
>
> > **Response to Q2.**
>
> - It's not a typo. The definition of "$k$-hop" we use here follows the definition from a paper accepted at NeurIPS 2022 [1]. Specifically, "$k$-hop neighbors of a node $v$" refers to all the neighbors that have a distance from node $v$ less than or equal to $k$," not exactly at a distance of $k$ from node $v$." If it was understood as the latter, then your statement of "$3$" is correct. To eliminate any ambiguity, we have added footnotes in the revised version to explain this.
>
>
> > **Response to Q3.**
>
> - The reviewer's conjecture is correct. In the context of the sentence mentioned by the reviewer, "$k$" indeed refers to the depth within the tree structure.
>
> - When we revisited the original text and reviewed the sentence in question, we noticed a minor issue with Eq. 3 below this section. We have made the necessary corrections and clarified the meaning of "$k$" in the revised version to enhance the reader's experience. Thank you for your assistance.
>
>
> > **Response to Q4.**
>
> - In Eq. 2 and the text immediately above it, we defined that for a general branch $b$, we can obtain its corresponding representation $\mathbf{r}\_{b}$ using an LSTM. The reviewer's concern regarding $\mathbf{r}\_{b\_{i}}$ in Eq. 3 is simply a specialization for a more specific branch $b\_{i}$. Here, we added an additional subscript because we needed to distinguish between a branch and its neighboring branches.
>
> > **Reference:**
>
> [1] Feng, Jiarui, et al. "How powerful are k-hop message passing graph neural networks." Advances in Neural Information Processing Systems 35 (2022): 4776-4790.

---

> ### Author Response · Authors · 2023-11-17
> **Response to Reviewer 7LXV (2/3)**
>
> > **Response to W1.**
>
> - We sincerely appreciate the reviewer's advice on the writing of our paper.
>     - In our previous manuscript submission, while we did not provide the specific definitions of metrics in the main text, we promptly directed readers to refer to the appendix for detailed metric definitions and explicitly indicated where in the appendix they could find this information. As shown in the specific metric definitions in the appendix, explaining these metrics in detail would require a considerable amount of space. Given that the main text needed to be limited to 9 pages, dedicating a substantial portion of it solely to introducing metric definitions would not have been appropriate. Furthermore, even though readers might not be familiar with the exact definitions of these metrics, we provided guidance on how to evaluate the performance of the results. For example, in the caption of Table 1, we explicitly mentioned, "A closer alignment with Reference indicates better performance."
>
>     - The section in our paper regarding the biological motivation/justification for local and global conditions immediately follows our introduction of extracting local and global conditions from previously generated layers. We believe that this arrangement provides a coherent and concise presentation. If we did not provide the biological motivation/justification at this point, readers might wonder why we decided to split the conditions into two categories and what the significance of each condition is. You mentioned that it would be best to place this content "after each method step". If you are referring to "method" as Sec 3.1, we believe our approach already meets this requirement. If you are referring to "method" as Sec 3.2, we believe that placing this content there would be less effective, as it would diminish the clarity of the motivation behind the two conditions mentioned Sec 3.1. Additionally, Sec 3.2 focuses on the practical instantiation of the method.
>
>     - Regarding your questions about the meaning of certain notations, we have already provided explanations for some of the notations in response to your previous questions regarding Q3 & Q4. Since our paper contains a considerable amount of notation, we have summarized the notations in the main text in Appendix K (for both the old and new versions of the paper). In the revised version, we have made some additional clarifications. If you have any further questions about other notations, please feel free to let us know, and we will promptly provide explanations.
>
>
>
> > **Response to W2.**
>
> - This is not a case of over-fitting. Please refer to our explanation below:
>    - From the original MorphVAE paper, it is evident that MorphVAE also requires a real sample as a reference when generating a new sample:
>         - The formulation of the neuronal morphology generation task in the original MorphVAE paper at Sec 2.4 is as follows: $$\hat{M}\_T=g_\phi\left(f_\theta\left(M_T\right)\right),$$
>           where $M_T$ is a set of 3D-walks that make up the morphology, which MorphVAE uses to represent a morphology. $f_\theta$ and $g_\phi$ represent the encoder and decoder, respectively. $\hat{M}_T$ represents the final generated 3D-walks.
>         - If there was no need for a reference during generation, there would not be a test dataset. However, the original MorphVAE paper clearly defines train-valid-test dataset splits.
>
>    - In our paper, we adopt the same experimental setting as MorphVAE. This setting is somewhat similar to "augmentation", and we have justified the validity of such a setting in the introduction of our paper (refer to the the fourth paragraph in Section 1 for both the old and revised versions).
>
>     - Since we, like the baseline MorphVAE, generate new neurons in an augmentation-like manner, it is justified that the generated morphology closely resembles the provided reference morphology, and this does not suggest over-fitting.

---

> ### Author Response · Authors · 2023-11-17
> **Response to Reviewer 7LXV (3/3)**
>
> > **Response to W3.**
>
>
>
> - Your approach may indeed achieve better performance on some metrics compared to our MorphGrower and baseline MorphVAE. However, we do not recommend generating more morphologies in this way unless a very reasonable perturbation method can be designed. The specific reasons are as follows:
>
>     - If you simply perturb branch points, it may result in abrupt changes in the branch, which should be avoided as much as possible. If such abrupt changes occur at the end of a branch, it may also affect the starting segment of subsequent branches.
>     - Perturbation design should also take into account the influence of predecessor branches on subsequent branches, i.e., how the perturbation of subsequent branches is conditioned on the perturbation of predecessor branches. It is also essential to consider that the perturbed branches should still follow the organizing principle of self-avoidance. As detailed in the revised paper's Appendix N.1, our global condition extraction module indeed has the ability to capture complex patterns in neuronal morphologies, thereby influencing subsequent branches. In contrast, the perturbation method, without careful design, cannot achieve this.
>     - Additionally, the strategy of removing some branches that you mentioned should also be approached with caution. This can affect the asymmetry of the tree structure, which is a factor to be considered in the process of neuron growth. Simply deleting some branches may lead to morphology with an unreasonable tree asymmetry metric.
>     - Furthermore, such a perturbation algorithm is sure to involve hyperparameter selection, and it would require reselecting suitable hyperparameters for different datasets. For instance, parameters controlling the degree of perturbation would need to be adjusted. The choice of these hyperparameters may heavily rely on expert experience. In contrast, our method is purely data-driven and can generalize more rapidly across various datasets.
>
> > **Response to W4.**
>
> - We have already provided an explanation for the first sentence mentioned by the reviewer in our response to Q1.
>
> - As for the second sentence, we acknowledge that there is an issue with its accuracy. Our intention here was to convey a relative concept, where dendrites' growth is slightly more random compared to soma, while axons exhibit more specificity. This sentence is not crucial to the main points, so we have chosen to remove it in the revised version.

---

> ### Comment · Reviewer_7LXV · 2023-11-21
>
> I appreciate the clarifications by the authors, and have gone through the revised manuscript.
>
> I agree with the authors that the main contribution is in generating perturbed versions of a given morphology (and not in generating de novo morphologies from the overall distribution of morphologies in a dataset). The proposed method is only used to produce and evaluate perturbed trees that are identical in terms of number and connectivity of branches.
>
> The method can also extract a global representation of a morphology; however there is no direct validation of the global representations obtained in this way. This could be an straightforward and interesting experiment in a future version of the paper, using labels e.g. the M1-EXC and M1-INH datasets.
>
> My central opposition to the paper stems from a persisting lack of clarity about why such morphology perturbations are needed / useful (independent of MorphVAE). The authors agree that a simple baseline augmentation strategy (see W3 and related response) can beat the proposed method on evaluations considered here. The manuscript lacks an evaluation to demonstrate the advantage of this method over such simple, reasonable augmentation strategies.
>
> I have retained my original score for these reasons.

---

> > ### Author Response · Authors · 2023-11-22
> > **Gratitude for Your Feedback**
> >
> > We deeply appreciate your response and the insightful suggestions offered. We are dedicated to meticulously enhancing our paper in light of your advice.

---

### Official Review · Reviewer_s2ri · 2023-11-01

**Soundness:** 3 good
**Presentation:** 3 good
**Contribution:** 4 excellent
**Rating:** 6
**Confidence:** 3

**Summary:**

The authors present MorphGrower, a model for neuron morphology generation based on reference morphologies as inputs. Their method features conceptual advances over the previous state of the art in the field, MorphVAE. A comprehensive evaluation suggests that these advances yield considerable benefit across a range of quantitative measures of performance.

**Strengths:**

MorphGrower constitutes a significant advance in learning-based neuronal shape generation compared to the pioneering MorphVAE. In particular, MorphVAE generates neurons by sampling soma-to-tip branches and agglomerating a set of such branches via threshold-based node merging. This can yield topologically infeasible morphologies; Furthermore, agglomeration by averaging node positions has a smoothing effect which is detrimental to the yielded shape variability.
To counter these deficiencies, MorphGrower generates shapes recursively, "layer by layer", where "layer" refers to branch distance from the soma (where a "branch" spans from bifurcation or soma to bifurcation or tip). By recursion, an encoding of the local path to a current branching point as well as an encoding of the full previous layer is fed as condition to a branch pair encoder and -decoder.

**Weaknesses:**

MorphVAE provides an embedding for whole neuronal morphologies (via pooling of walk embeddings), which can be leveraged for shape clustering and cell type classification. This feature is not straightforwardly contained in MorphGrower as the respective encoder operates recursively on neuron branches. While MorphGrower is clearly pitched as focusing on neuronal shape generation, an explicit discussion of the aforementioned distinction in scope from MorphVAE would still be helpful for the reader.

While the provided evaluation of neuronal shape generation is comprehensive and shows clear benefits of MorphGrower over MorphVAE, it would still be beneficial to also report the shape characteristics statistics employed in MorphVAE (cf. their Fig. 5). Furthermore, it appears that MorphVAE has been re-trained by the authors with hyper parameters different from the original model, which are then however applied to (partly) the same date -- would it be possible to directly use the resp. models trained by the MorphVAE authors, or at least their exact hyperparameters?

Further details:

For the branch pair decoder, it would be helpful if you could discuss respective permutation equivariance -- do you include both orders of each branch pair during training to train towards equivariance? or is the architecture inherently permutation equivariant? (if so this is not straightforwardly obvious)

In your comparative evaluation vs MorphVAE, it would be beneficial if you could provide a more comprehensive discussion of hypotheses regarding the sources of the observed differences. E.g., MorphVAE caps walk lengths, which clearly entails an underestimate of some of the measures you evaluate, yet this source of underperformance is not discussed.

**Questions:**

Would it be possible to directly use the models (or at least hyper parameters) employed by the MorphVAE authors, at least for your comparative evaluation on data also used in the MorphVAE work?

Could you extend your evaluation of morphological statistics to the measures evaluated in the MorphVAE work?

Furthermore, please explicitly discuss the distinct scope of MorphGrower vs MorphVAE.

---

> ### Author Response · Authors · 2023-11-17
> **Response to Reviewer s2ri (1/2)**
>
> Thank you for your insightful comments. Below, we address your concerns with our responses.
>
> > **Using MorphVAE's authors' provided pre-trained model or hyperparameters on the dataset shared by us and MorphVAE.**
>
> - We could not directly utilize the models or hyperparameters provided by the MorphVAE authors for the following specific reasons:
>
>     - The baseline MorphVAE did not provide a pretrained model in their open-source GitHub repository. Therefore, we had to train MorphVAE ourselves.
>     - The authors of MorphVAE provided datasets, but these datasets were in a processed format that could be directly fed into the MorphVAE model, which was not suitable for our model. Consequently, we sought out the original datasets for these three datasets and discovered that, for example, in the case of the RGC dataset, the original dataset contained more samples than the RGC samples used by MorphVAE. Mismatched sample counts implied inconsistent data splits, making it inappropriate for us to directly adopt the hyperparameter choices from the MorphVAE original paper. We had to perform a new search for hyperparameters.
>     - During our grid search for training MorphVAE's hyperparameters, we included the training hyperparameter settings provided by the MorphVAE original paper. Additionally, we strictly followed MorphVAE's resampling distance parameter for data preprocessing, which was the same for all three datasets used by both us and MorphVAE (we mentioned it in Appendix H.1 in both the old and new versions of the paper).
>
> > **Evaluating the effectiveness of MorphGrower using metrics adopted in MorphVAE.**
>
>
> - Based on the feedback from the reviewer, it is evident that the reviewer conducted a thorough and responsible review, including a review of the original paper on baseline MorphVAE. We deeply appreciate this aspect of their review.
>
> - Both baseline MorphVAE and our MorphGrower require a real sample as input, referred to as a "reference" in the paper, to generate new samples that closely resemble the input. MorphVAE employs a 3D-walk as the fundamental generation unit, followed by clustering to obtain the final morphology. This approach can result in discrepancies between the tree structure of the generated morphology and the reference, and may even lead to the creation of invalid branching points (with more than two successor nodes).
>
> - In contrast, we utilize a layer-by-layer generation strategy and generate branches in pairs within each layer. This approach ensures that the tree structure of the final morphology we generate strictly aligns with the reference.
>
> - You mentioned that the original MorphVAE paper's Figure 5(b) includes six metrics. Among these, four metrics—max branch order, tree asymmetry, width, and depth—are employed to assess the similarity of the generated morphology's tree structure to the provided reference. Therefore, these four metrics are guaranteed to be identical to the reference and are thus superior to baseline MorphVAE. Consequently, we believed that presenting the results for these metrics was not essential. The remaining two metrics, mean soma exit angle and mean branch angle, are angle-related, and in our previous manuscript submission, we have introduced two better-defined angle metrics and presented their results.

---

> > ### Comment · Reviewer_s2ri · 2023-11-21
> >
> > Thank you for your comprehensive response. It clarifies that the best effort was made to reproduce MorphVAE's hyper parameters and model. Did you perform a sanity check of your reproduction of the MorphVAE model by means of their evaluation metrics to see if these are (approximately) reproduced as well? Or would this not be meaningful due to the flaws in the data splits you encountered?\
> > Your response also clarifies that four of the six MorphVAE evaluation metrics do not make sense for MorphGrower as the latter guarantees identical topology. Could you please still elaborate on the "better-defined angle metrics" you mention? How do they improve upon the respective MorphVAE metrics?
> > Many thanks!

---

> ### Author Response · Authors · 2023-11-17
> **Response to Reviewer s2ri (2/2)**
>
> > **The distinct scope of MorphGrower vs MorphVAE.**
>
>
> - MorphVAE primarily addresses two main issues:
>
>     - It introduces method for augmenting neuronal morphology data using 3D walks as the fundamental generation unit.
>
>     - It proposes a method for extracting an overall representation of neuronal morphology data: a pooling operation is applied to the latent embeddings corresponding to all walks. This overall representation can be applied to downstream tasks such as neuron classification.
>
>    In MorphVAE, the generation module and the downstream classifier are trained simultaneously. However, it's worth noting that MorphVAE's primary focus and contribution lie in augmenting neuronal morphology data. The extraction of overall features of neuronal morphology, as described in the second point, can be considered an additional byproduct. Additionally, MorphVAE uses a relatively large step size for resampling neuron data, resulting in the loss of a substantial amount of fine-grained information and the possibility of introducing multiple branching points in the augmented data. This does not align with the reasonable requirements of neuronal morphology topology, further indicating that MorphVAE may not be a particularly suitable model for augmenting neuronal morphology data.
>
> - MorphGrower focuses on augmenting neuronal morphology data at a finer granularity: We employ a layer-by-layer generation strategy and generate branches in pairs within each layer. We do not aim to learn an overall representation of neuronal morphology for downstream tasks. However, it is worth noting that we have found that the global condition module proposed by MorphGrower also has the ability to capture complex patterns in the data, as detailed in revised paper's Appendix N.1. Additionally, we use a more reasonable data preprocessing approach by using a smaller step size for normalizing neuron samples. These measures ensure that our augmentation pipeline results in neuron data with more detail and guarantees the topological validity of generated samples.
>
> - We appreciate the reviewer's suggestion, and we have provided a discussion of the differences between the two approaches in revised paper's Appendix L.
>
>
> > **Question about the both orders of each branch pair during training.**
>
> - During training, we do not fix the order of the two branches in each branch pair; their order is randomized in each epoch. Therefore, during training, we include both possible orders of the two branches within the same branch pair, enabling the model to train towards permutation equivariance.
>
>
> > **More comprehensive discussion of hypotheses regarding the sources of the observed differences.**
>
> - The reviewer's suggestions are greatly appreciated. In response, we have added more discussion on hypotheses regarding the sources of the observed differences in the revised manuscript, which has been resubmitted.

---

> > ### Comment · Reviewer_s2ri · 2023-11-21
> >
> > I appreciate that you've included an explicit discussion of MorphVAE's vs MorphGrower's scope, as well as an extended discussion of hypotheses regarding the observed differences, in the revised manuscript. Last but not least, my permutation equivariance question is fully clarified.

---

> > > ### Comment · Reviewer_s2ri · 2023-11-22
> > >
> > > I agree with the other Reviewers that it would be great to show the benefits of the neuronal morphology augmentation provided by MorphGrower in downstream (learning) tasks. However, the authors convincingly show that their augmentations are more realistic than the previous state of the art, which suggests good potential for use in downstream tasks. Thus hence I stick to my original score.

---

> > > ### Author Response · Authors · 2023-11-22
> > > **Appreciation for Your Insightful Comments**
> > >
> > > First of all, thank you very much for your response. Here is our detailed reply to your remaining concerns:
> > > > **Reproducing MorphVAE.**
> > >
> > > You are correct in your thinking that the inconsistency in data split is indeed a reason that makes it challenging to check the reproduction of the MorphVAE model.
> > >
> > > > **Our better-defined angle metrics.**
> > >
> > > The angles in MorphVAE are defined at the compartment level (as detailed in Definition 2 in Section 2 of our paper regarding the definition of a compartment). Specifically, the angles defined in MorphVAE are based on the angle formed by two compartments that overlap at one end. Therefore, compared to the two angle-related metrics defined in our paper, this is not only heavily influenced by the number of points in each branch after resampling, but it also does not effectively reflect the overall angular relationship between branches.
> > >
> > > We hope our response effectively addresses your concerns.
> > >
> > > We sincerely thank you for your response and suggestions, and we will continue to work hard to improve our work.

---

### Author Response · Authors · 2023-11-17
**General Response**

We express our gratitude to all the reviewers for dedicating their time and providing valuable comments. They acknowledged that our work is well-motivated (NzbM), well-presented (s2ri, NzbM), meaningful (s2ri,  7LXV), includes comprehensive experiments (s2ri, NzbM), and demonstrates a significant improvement compared to the baseline MorphVAE (s2ri, NzbM). In particular, we are particularly thankful that all the reviewers recognized and endorsed our approach of progressively generating morphologies.

In the following response, we provide detailed answers to all the questions and comments point-by-point to further strengthen our contributions. We deeply appreciate the suggestions to improve this paper, and we have made corresponding modifications to the manuscript and have resubmitted a revised version.

If you have any further questions, please let us know so that we can provide a timely follow-up response.

---

### Author Response · Authors · 2023-11-20
**Sincerely Awaiting Your Feedback - Discussion Phase Coming to a Close**

Dear Reviewers,

We are deeply appreciative of the time and effort you have devoted to reviewing our paper. The insights you've shared have been crucial in helping us refine our work. As the discussion phase nears its conclusion, we truly hope that our detailed responses have met your expectations and assuaged any concerns. If there remain unresolved questions, we are more than willing to provide any necessary clarifications. Should our explanations bring clarity, it would be an honor for us if you might reconsider the evaluation of our work.

With heartfelt gratitude and warmest regards,

The Authors

---

### Author Response · Authors · 2023-11-20
**Wholeheartedly Awaiting Your Feedback**

Dear Esteemed Reviewers,

We wish to begin by expressing our deepest appreciation for the dedication and meticulousness you have shown in reviewing our manuscript. The depth of your insights and the constructiveness of your feedback have been truly invaluable. Your contributions have not only improved our work but have also provided us with new perspectives to consider.

We understand that this year's review process differs from previous years, notably in the absence of a second round of discussions between authors and reviewers. With the discussion deadline fast approaching on November 22, 2023, we realize that our opportunities for further dialogue are limited.

We have submitted detailed responses to address the concerns you raised. We have endeavored to cover every aspect with the utmost attention to detail, hoping that our clarifications meet your expectations. However, if there are any aspects that require further elucidation or if new questions have arisen, please feel free to contact us. We value your insights immensely and are fully committed to engaging in any additional conversations necessary to ensure absolute clarity and understanding.

Once again, we extend our heartfelt thanks for your invaluable contribution to our work. We eagerly await any further feedback you may have and remain dedicated to refining our manuscript to its fullest potential.

With warm regards and the highest respect,

The Authors

---

### Meta-Review · Area_Chair_ummo · 2023-12-13

**Metareview:**

The paper proposes a new method for generating perturbed neuronal morphologies (augmentations) from an existing population of template morphologies. This was a borderline paper. The reviewers agreed that the method represents an advance over MorphVAE, a previous approach. However, they expressed concerns regarding the ultimate utility of this method.

**Justification For Why Not Higher Score:**

Concerns regarding the utility of the proposed method, of limited interest.

**Justification For Why Not Lower Score:**

NA

---

### Decision · Program_Chairs · 2024-01-16

Reject